# Applying double-cropping and interactive irrigation in the North China Plain using WRF4.5

Yuwen Fan[1], Zhao Yang[2], Min-hui Lo[3], Jina Hur[4], Eun-soon Im[1,5]

[1]Division of Environment and Sustainability, The Hong Kong University of Science and Technology, Hong Kong, China
[2]Pacific Northwest National Laboratory, Richland, WA, USA
[3]Department of Atmospheric Sciences, National Taiwan University, Taipei, Taiwan
[4]National Institute of Agricultural Sciences, Rural Development Administration, Wanju-gun, Jeollabuk-do, Korea
[5]Department of Civil and Environmental Engineering, The Hong Kong University of Science and Technology, Hong Kong, China

*Correspondence to*: Yuwen Fan (yfanaj@connect.ust.hk), Eun-soon Im (ceim@ust.hk)

**Abstract.**

Irrigated cultivation exerts a significant influence on the local climate and the hydrological cycle. The North China Plain (NCP) is known for its intricate agricultural system, marked by expansive cropland, high productivity, compact rotation, a

semi-arid climate, and intensive irrigation practices. As a result, there has been considerable attention on the potential impact of this intensive irrigated agriculture on the local climate. However, studying the irrigation impact in this region has been challenging due to the lack of an accurate simulation in crop phenology and irrigation practices within the climate model. By incorporating double-cropping with interactive irrigation, our study extends the capabilities of the Weather Research Forecast (WRF) model, which has previously demonstrated commendable performance in simulating single-cropping

scenarios. This allows for two-way feedback between irrigated crops and climate, further enabling the inclusion of irrigation feedback from both ground and vegetation perspectives. The improved crop modeling system shows significant enhancement in capturing vegetation and irrigation patterns, which is evidenced by its ability to identify crop stages, estimate field biomass, predict crop yield, and project monthly leaf area index. The improved simulation for large-scale irrigated crop in the NCP can further enhance our understanding of the intricate relationship between agricultural

development and climate change.

**Plain Language Summary.**

Irrigated agriculture in the North China Plain (NCP) has a significant impact on the local climate, but the existing climate models do not accurately simulate the crop and irrigation. To address this limitation, we add a double-cropping function to

Weather Research Forecast model. We also recalibrate the parameters to simulate the crop growth and irrigation amounts more accurately. Our improved model better captures crop calendar, biomass, vegetation fraction, as well as monthly leaf area index.

## 1 Introduction

Agriculture serves as one of the primary drivers of land use changes (Goldewijk, 2001) and the largest consumer of water
resources globally (Foley et al., 2011). To increase crop productivity and feed the exploding population, irrigation has
rapidly expanded in the past decades, and accounts for over 70% of the global freshwater withdrawal today. This intensive
and extensive irrigation undoubtedly exerts a significant influence on the hydroclimate (McDermid et al., 2023; Siebert et
al., 2010). While it is widely acknowledged that irrigation has a cooling and moistening effect on a global scale (Cook et al.,
2011; Lo et al., 2021; Pokhrel et al., 2012; Puma and Cook, 2010), its influence is non-linear and location-specific at
regional scales, as it greatly depends on the agricultural and climatic conditions of the region in which it is deployed (Fan et
al., 2023; Im et al., 2014; Kang and Eltahir, 2018, 2019; Pei et al., 2016; Tuinenburg et al., 2014; Wey et al., 2015; Yang et
al., 2019). Consequently, these complex and unpredictable changes induced by irrigation have attracted considerable
attention, underscoring the need to improve crop representation and effectively simulate the interactions between irrigated
cultivation and regional climate.

Numerous studies have simulated the irrigated crops using traditional agricultural models (DeJonge et al., 2012; Menefee et
al., 2021) or offline land surface models (Lombardozzi et al., 2020; Yin et al., 2020). However, while the vegetation patterns
and irrigation practices gradually alter the climatic processes, the changing climate also influences back onto crop growth
(Ahmed et al., 2015; Choi et al., 2017; Pielke et al., 2007; Ramankutty et al., 2006; Yang and Wang, 2023). This two-way
interactive feedback between irrigated agriculture and climate can only be captured when employing an interactive crop
system within the climate models (Chen and Xie, 2011; Harding et al., 2015; Lu et al., 2015). These interactive crop models
can not only capture the temporal pattern of crop growth but also depict spatial heterogeneity at regional scales with
relatively fast computational speed (Chen and Xie, 2011; Liu et al., 2016; Oleson et al., 2013; Yin and van Laar, 2005).
When simulating the water forcings that sustain crop growth, some models simply assume no irrigation (Liu et al., 2016),
while others incorporate irrigation with fixed amount (Vira et al., 2019) or dynamically adjust the irrigation amount based on
daily soil conditions (Ozdogan et al., 2010; Qian et al., 2013; Valayamkunnath et al., 2021; Wu et al., 2018b; Yang et al.,
2016, 2017, 2019, 2020). With these algorithms to simulate crop phenology and irrigation behaviour, multiple studies have
reported significant enhancements in dynamic vegetation predictions and a better understanding of irrigation impact (Xu et
al., 2019; Yang et al., 2016; Zhang et al., 2020).

However, irrigated agriculture has not been explicitly represented in most regional climate models. One key issue is the
inadequate coupling between the crop module and the irrigation module. For instance, many studies adopt prescribed
vegetation, which means that the crop growth may not be sensitive to the water forcings (Lu et al., 2015). Also, the irrigation
activation is often prescribed by date rather than following the actual crop season. The missing connection between crop and
irrigation introduces uncertainties in capturing the climatic processes, as both crop physiology and climate variations

dynamically influence each other (Fang et al., 2001; Porter and Semenov, 2005). The second issue is the applicability. Global-scale datasets related to cropland factors have not kept pace with other vegetation mappings (Oleson et al., 2013), and thus, schemes are predominantly developed and calibrated in field scale in the United States, which are mostly rainfed cropland (Menefee et al., 2021). Therefore, regionalizing the model and improving their adaptation for large-scale irrigation over other parts of the world becomes imperative.

Previous studies have shown that the regionalization process significantly improves the model performance. This process includes not only parameter calibration (Hong et al., 2015; Liu et al., 2010; Park and Park, 2021; Xie et al., 2007) but also algorithm modifications to enhance the model's applicability to different regions (Bou-Zeid et al., 2007; Livneh and Lettenmaier, 2013; Song et al., 2022). For instance, recalibration has been shown to significantly enhance crop prediction accuracy in Northeast China and southwestern Europe (Asmus et al., 2023; Yu et al., 2022). Introducing new tuning factors into the default equation can aid in simulating unique vegetation patterns within specific study domains (Wu et al., 2018b). Upgrading a variable such as the irrigation threshold from a single constant to a spatially varied 2D variable can better capture the spatial variability of irrigation application (Xu et al., 2019; Zhang et al., 2020). Additionally, incorporating new irrigation methods for paddy cropland improved irrigation predictions for southern Asia (Yao et al., 2022). These enhancements underscore the importance and efficacy of regionalization in improving the simulation in irrigated agriculture.

As a key agricultural region, the North China Plain (NCP) encompasses more than 40% of China's total harvested area (FAO, 2019). Approximately two-thirds of the land within the NCP is dedicated to cropland, contributing to nearly half of the nation's wheat production and one-third of the corn production (Wang et al., 2008). However, the annual precipitation in the NCP is only around 800mm, which is nearly half of that in southern China (Zhe et al., 2014), increasing its dependency on irrigation. Spatially, approximately 40% of the farmland in the NCP relies on irrigation (Portmann et al., 2010; Siebert et al., 2013). The significant effects of irrigation on the relatively dry climate in the NCP have been demonstrated (e.g. Fan et al., 2023). Thus, the NCP is an ideal testbed for studying irrigated crops and climate feedback, rooting not only in its extensive cropland and high productivity but also in its semi-arid background, and intense irrigation. This specific crop rotation exerts a profound influence on vegetation patterns and irrigation requirements, consequently leading to notable regional climate modifications (Jiang et al., 2021). This crop rotation greatly affects the vegetation pattern and irrigation demand, further alter the regional climate (Jeong et al., 2014). Furthermore, the spring irrigation, which is supplied for winter-season cropping during relatively dry season, can have a particularly pronounced impact on the local climate (Fan et al., 2023; Wu et al., 2018b). However, most current crop models in land surface models (LSMs) primarily account for single cropping. Therefore, it is necessary to consider this distinctive double-cropping rotation, along with other local characteristics, to accurately capture the crop growth and irrigation activities.

Given the unique characteristics of the NCP, our research aims to simulate the irrigated crop growth with a double-cropping rotation, which is specifically tailored for the NCP and its surrounding region. To achieve this, Noah land surface model with multiparameterization options (Noah-MP) (Niu et al., 2011) has been selected, as it already encompasses several functions related to cultivation simulation, and has consistently exhibited exemplary performance in previous studies when simulating single-cropping scenarios (Liu et al., 2016; Xu et al., 2019; Zhang et al., 2020). Its crop model is already implemented within the Weather Research and Forecasting Model (WRF) (Skamarock et al., 2019) to enable two-way nested feedback between the crop system and climate dynamics. While conducting parameter calibration and adopting local inputs to capture more local details, we also try to integrate satellite data to assess its ability in large-scale simulation. By integrating and regionalizing the crop modeling system, this study primarily focuses on the model development and its predictability assessment in crop phenology and irrigation requirements, which represents a promising avenue for advancing our understanding of the coupled human-natural system. The incorporation of satellite input also holds the potential to enhance the applicability of our approach in various regions beyond the current study area.

## 2 Model Description and Experiment Design

The study domain is centered on the NCP, encompassing a significant portion of China's cropland. Given the unique characteristics of this region, we anticipate that the model will exhibit the following capabilities:

- Accurate representation of the general vegetation and irrigation patterns in the NCP region, especially the presence of double crop seasons.
- Integration of direct interactions between crops, irrigation, and climate, with sensitivity of each factor to the other two. In other words, the model should account for the influence of crop growth and irrigation practices on the local climate, while also considering the impact of climate conditions on crop development and irrigation requirements.

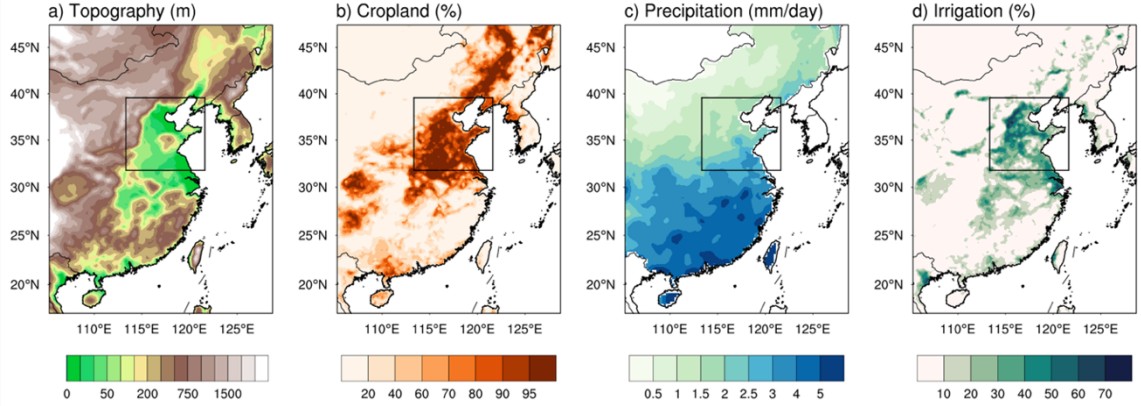

**Figure 1: (a) Topography (m), (b) cropland fraction (%), (c) annual precipitation (mm/day), and (d) irrigated land fraction (%).**

## 2.1 Study Area

Figure 1 illustrates some key background variables, outlining the NCP region within black boxes. The topography and cropland fraction are basic geostatic inputs for the WRF, initially retrieved from the United States Geological Survey and Moderate-resolution Imaging Spectroradiometer (MODIS), respectively. Notably, the NCP region, being the largest plain in eastern China, exhibits an average elevation below 100m (Fig. 1a), contributing to its suitability for cultivation. Despite the high cropland fraction exceeding 95% in most of the pluvial area (Fig. 1b), the climatology annual precipitation (retrieved from China Meteorological Forcing Dataset) in 2000-2009 is merely half that of southern China (Fig. 1c), highlighting the need for irrigation. According to the FAO AQUASTAT database (Siebert et al., 2013), irrigated cropland constituted more than 70% of the total land use in the pluvial area in 2005 (Fig. 1d).

## 2.2 Model Configuration and Experiment Design

The study employs the Advanced Research version of the WRF Model (version 4.5), a non-hydrostatic numerical weather prediction model that has been widely adopted in regional studies. Model domain is shown in Fig. 1. This study only employs a single domain which is depicted as the entire map in Fig. 1, while the inner black box in Fig. 1 serves solely for the identification of the NCP region. The horizontal grid spacing is 27km, with 38 vertical layers in the atmosphere and 4 soil layers below the ground. Its physical options mostly follow Fan et al. (2023), including the WRF double-moment 5-class microphysical parameterization (Hong et al., 2004), the Rapid Radiative Transfer Model as the longwave radiation scheme (Mlawer et al., 1997), the Dudhia shortwave radiation scheme (Dudhia, 1989), the Yonsei University planetary boundary layer scheme (Hong et al., 2006), the scale-aware New Simplified Arakawa-Schubert scheme (Han and Pan, 2011; Kwon and Hong, 2017), and Noah-MP land surface model coupled with our improved crop and irrigation schemes (Ek et al., 2003). The initial and lateral boundary conditions are obtained from the 6-hourly ERA-Interim reanalysis dataset, which helps to reduce the uncertainty arising from the boundary condition (Dee et al., 2011).

**Table 1. Description of all WRF simulations**

| WRF simulation | Model | | |
|---|---|---|---|
| | Vegetation | Crop | Irrigation |
| CTL | Prescribed Input | | |
| CROPdef | Predicted by crop model | default version | |
| CROPnew | Predicted by crop model | improved version | |
| IRRdef | Predicted by crop model | improved version | default version |
| IRRnew | Predicted by crop model | improved version | improved version |

We commence by calibrating the crop growth and irrigation behaviour in 2005, representing normal conditions based on the East Asian Summer Monsoon Index (following the definition from Li & Zeng, 2002). To account for the typical sowing of winter wheat in the autumn of the preceding year, all simulations are initiated on 1st March 2004. This allows for a spinning-up period of at least six months before the 2004-2005 crop season, ensuring that the model was appropriately initialized for accurate simulations. Subsequently, a ten-year period spanning from 2005 to 2014 is employed for validation, utilizing long-

term data to assess the overall performance and the stability of both crop prediction and irrigation simulation, respectively.

Detailed information regarding all WRF simulations can be found in Table 1, which provides a detailed description of how vegetation, crops, and irrigation are simulated in our study. All models are inactive in the control experiment (CTL), in which static vegetation with predefined monthly patterns from satellite data is employed. The crop and irrigation model can

be applied either in the default version or the improved version. The default crop model is conducted using the original scheme proposed by Liu et al. (2016) and parameters derived from Zhang et al. (2020), while the improved crop model involved both modifications to the algorithms and recalibration of the parameters. In order to exclusively demonstrate the advancements made by the crop model, the irrigation component remains inactive in both CROPdef and CROPnew. This implies that no supplementary water is introduced to the cropland, thereby highlighting the impact solely attributed to the

crop model. The added value of our improvements on the irrigation model can be discerned through a straightforward comparison between IRRdef and IRRnew simulations. In IRRdef, the default version of dynamic irrigation is derived from He et al. (2023) and serves as the baseline for the improved version. In the default version, the target soil moisture availability as a parameter threshold is uniformly set to 0.8, as suggested by Fan et al. (2023), while in the improved version, it exhibited spatial variability between provinces. The detailed improvements made to the crop and irrigation models will be

explained in Sections 2.3, and 2.4, respectively.

## 2.3 Modification of the crop model

### 2.3.1 Crop area and FVEG prediction

In order to achieve efficient computation, the crop module developed by Liu et al. (2016) is selected as the foundation for

crop simulation. This particular crop model is initially designed for crop fields and thus applied uniformly to all the grids within the domain. However, to extend its application to a larger domain that has various land-use types, the model needs to be exclusively activated on crop grids, while non-crop grids still utilize prescribed vegetation as the CTL. A crop grid is defined based on MODIS land-use classification as either 'Croplands' or 'Cropland/Natural Vegetation Mosaic'. This definition aligns with Fan et al. (2023), and is similar to the approach employed by Yu et al. (2022) who set a threshold of

50% cropland percentage, since the majority of grids in the NCP region contain over 90% cropland (Fig. 1b).

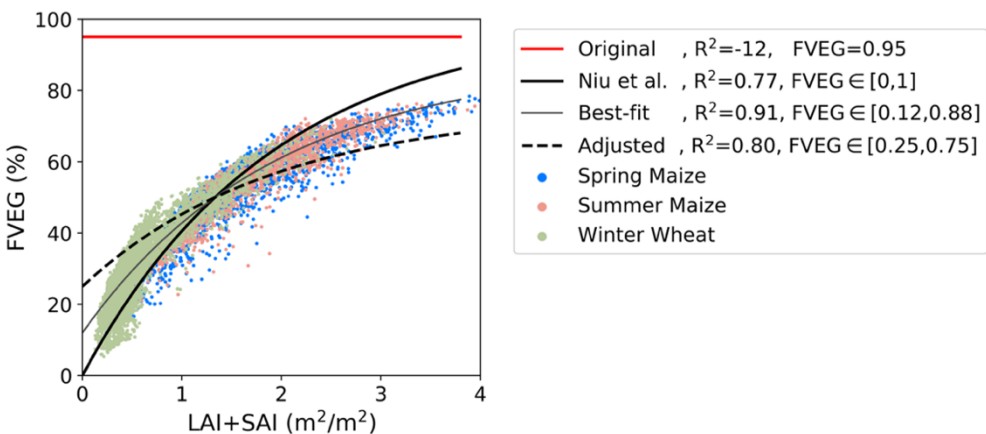

**Figure 2. Satellite-based daily FVEG (vegetation fraction) and LAI+SAI (sum of leaf area index and stem area index) over the crop season are represented by colorful dots for each grid in NCP. Different dot colors indicate different crop types. The lines display the relationships that we attempted to simulate the FVEG based on LAI+SAI.**

Although the dynamic leaf area index (LAI) and stem area index (SAI) can be calculated based on crop growth and climate conditions, the default crop model simply assumes the vegetation fraction (FVEG) to be 95% for all grids at all times (red line in Fig. 2), to represent the dense vegetation in the crop field. However, this fixed value is not appropriate for regional-scale applications. FVEG is a fractional factor that determines the proportion of solar radiation captured by the canopy, as well as the contribution of vegetation to the ground-released energy. Considering the long-term impact of vegetative

radiation and canopy interception (Liu et al., 2020; Wang et al., 2007), FVEG should be correlated with the vegetation growth with spatial and seasonal variation. Therefore, we first try to correlate the FVEG with LAI/SAI using the empirical relationships (shown in Equation 1 and the thick black line in Fig. 2). This equation is proposed by Niu et al. (2011) and further testified by Wu et al. (2018b) in the NCP region. However, according to the MODIS observation retrieved from the input of the CTL, it is imperative to note that the original curve underestimates the FVEG at low LAI+SAI and overestimates

it at high LAI+SAI, which poses a potential risk to the reliability of the predictions. More specifically, at the onset of the crop season (when LAI+SAI is small), accurate LAI+SAI estimation leads to an underestimation of the calculated FVEG. This, in turn, results in reduced shortwave radiation intercepted by vegetation and a slower rate of photosynthesis. Consequently, the leaf growth is undervalued in the next timestep, and the less LAI creates a larger bias on the FVEG prediction. This positive feedback continues to accumulate underestimation during subsequent iterations, and ultimately,

results in the failure of the entire crop season. Similarly, the curve exhibits an exaggerated FVEG during the flourishing period (when LAI+SAI is large), which easily leads to uncontrollable overgrowth. This susceptibility underscores the necessity to consider and address this inherent limitation. Even when employing the best-fitting curve, this issue persists for almost half of the grids (for those who have greater FVEG at low LAI+SAI or lower FVEG at high LAI+SAI). Therefore, we finally adopt the adjusted line by proposing a constraint on the range of FVEG, limiting it to [0.25, 0.75], instead of

utilizing the full range of [0, 1]. This allows for a slight overestimation in the initial stages and an underestimation towards the end, ensuring a successful startup and a steady progression toward its peak. The adjustment on this equation enables the spatial and temporal variations of FVEG, as well as the vegetation responses to the irrigation application. Quantitatively, the adjusted curve demonstrates improved performance compared to the one extracted by Niu et al. (2011), achieving an R-square score of approximately 0.8, suggesting a commendable fit of the adjusted curve. It is worth noting that this validation focuses solely on the crop season in NCP. When adopting this crop model in other regions, a re-calibration would be required to ensure that the equation exhibits a slight overestimation during the initial stages and an underestimation towards the later stages of crop growth. Equations (1) and (2) below represent the original FVEG equation by Niu et al. (2011) and the adjusted FVEG suggested in this study, respectively:

$$\text{Original FVEG} = 1 - e^{(-0.52 \times (\text{LAI} + \text{SAI}))}, \text{FVEG} \in [0,1] \tag{1}$$

$$\text{Adjusted FVEG} = 0.75 - 0.5 \times e^{(-0.52 \times (\text{LAI} + \text{SAI}))}, \text{FVEG} \in [0.25, 0.75] \tag{2}$$

### 2.3.2 From single cropping to double cropping

The default model only considers single cropping, allowing different crops spatially but only one crop type per grid. However, NCP widely adopts double-cropping rotation, as evident from satellite vegetation patterns (Qiu et al., 2022; Wu et al., 2010; Yan et al., 2014; Yuan et al., 2020). The first growing season typically begins in late spring to early summer and extends until mid to late autumn, followed immediately by the second growing season which stops just before the restart of the first growing season. And it's necessary to consider the second crop season in the irrigated crop system, because the dry soil in the winter and spring probably requires a significant irrigation supply (Fan et al., 2023; Koch et al., 2020; Wu et al., 2018b; Yang et al., 2016). According to the crop prevalence (Qiu et al., 2022; Wu et al., 2010), we select winter wheat and summer maize for double cropping region (shown in orange in Fig. 3a), as identified by satellite data (Qiu et al., 2022), and spring maize for single cropping region (shown in blue in Fig. 3a).

The planting and harvesting dates are fed into the crop model to define crop seasons, whose spatial variability is claimed to be beneficial to the accuracy of crop growth prediction (Xu et al., 2019; Zhang et al., 2020). The harvesting date of the spring maize is assigned to be 15 days after the physiological maturity date obtained from a satellite-based post-processed dataset (Luo et al., 2020). The planting date is determined as 15 days prior to the V3 stage, which represents the early vegetative stage of maize when the third leaf is fully expanded. Similarly, for double-cropping regions, the maturity dates of wheat and maize, with a 15-day buffer, mark the end of the respective cropping seasons, while the subsequent cropping season starts 5 days later. The '15-day' buffer and '5-day' interval are roughly defined according to the LAI pattern in Luo et al. (2020). Few grids not covered by the satellite dataset are assigned 1 May (121st Julian Day) and 11 October (284th Julian

Day) as the default planting and harvesting date for maize, respectively, based on field study (Yu et al., 2022). The planting date and the harvesting date also perform similar spatial patterns to those generated by Wu et al. (2010).

### 2.3.3 Input Setting and Parameter Calibration

Starting with the parameters for one-year corn in Bondville (Zhang et al., 2020), we chose the Yucheng (36.83°N, 116.57°E) and Shenyang (41.52°N, 123.39°E) stations for calibration because of their availability of long-term data (provided by National Ecosystem Science Data Center, National Science & Technology Infrastructure of China). As shown in Fig. 3a, Yucheng represents a double-cropping system as a typical representation of the NCP region, while Shenyang, located nearby, represents a single-cropping system. The availability of long-term data at these stations ensures the reliability and robustness of our calibration process.

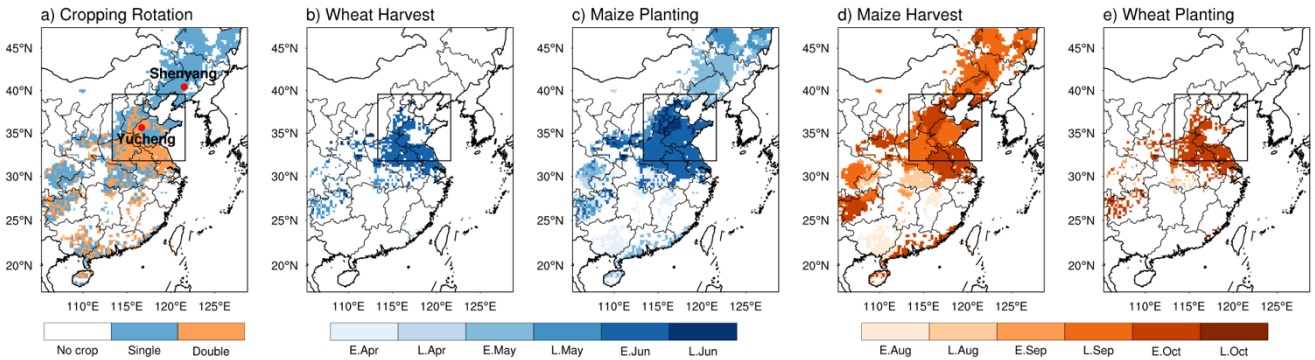

**Figure 3. Spatial distribution of (a) the cropping system with two stations used for calibration, (b-e) harvest date and planting date for wheat and maize over a year based on the chronological order. 'E. Apr' and 'L. Apr' is the abbreviation for Early and Late April.**

Based on the defined cropping area, the planting and harvesting dates are determined using the method outlined in section 2.3.2. The chronological sequence of these dates is presented in Figs. 3b-e. In regions with a single cropping system, spring maize is typically planted in May and harvested in September. On the other hand, in those double-cropping regions, winter wheat is usually harvested in late May or early June, immediately followed by the planting of summer maize. Next, maize harvest generally takes place in late September or early October, again followed by the planting of winter wheat, which continues to grow until the next year.

In the case of spring maize and summer maize, we first try to adopt the parameters from previous studies to keep the generality, and only recalibrate if necessary. For instance, large regional uncertainties may exist in the rubisco capacity (Vcmx25) and the leaf area per living leaf biomass (BIO2LAI) for summer maize (Yu et al., 2022; Zhang et al., 2020),

which probably require local validation. Conversely, a new set of parameters is developed specifically for winter wheat, drawing upon statistical information from the Yucheng station, satellite datasets, and other agronomy studies (Zhang et al., 1991, 2023). More specifically, the calibration for winter wheat includes the identification of crop stage, the calculation of general growth rate and the establishment of carbohydrate allocation. It is important to highlight that the calibration process was specifically carried out with the incorporation of updated irrigation algorithms, because the high productivity observed in the NCP is predominantly supported by irrigation in reality. Table S1 provides the adjusted parameters for wheat and maize, along with the supporting scientific references. Parameters are initially recalibrated in Yucheng and Shenyang using station data. Subsequently, these parameters are applied to the whole domain, with validation of vegetation pattern (i.e., LAI, FVEG, grain mass and crop calendar) conducted to ensure their spatial applicability to the whole region.

The recalibration starts from crop-stage identification, since it relies purely on the accumulated GDD and is less affected by other crop parameters. The GDD-related parameters are retrieved from Zhang et al. (2020) and Zhang et al. (1991), and then validated with the heading date and maturity date retrieved from the satellite data (Luo et al., 2020). The crop stage comprises the pre-planting stage, three vegetative stages (emergence, initial vegetative, post-vegetative), two reproductive stages (initial reproductive, post-reproductive), and finally, one maturity stage. During the vegetative stage, a majority of carbohydrates are allocated to the leaves and stems, while in the reproductive stage, the allocation shifts towards the grain.

Next, the general growth rate including BIO2LAI can be extracted from the station data, and Vcmx25 can also be estimated using the monthly satellite data of gross primary product (GPP) and LAI, since the photosynthesis rate and the LAI can be considered linearly related, especially on sunny days when the canopy temperature is around 25°C (He et al., 2023). The GPP and LAI that we adopted for validation are initially derived from MODIS products but have undergone further post-processing to generate a more continuous monthly pattern (Wang et al., 2020; Yuan et al., 2020). Furthermore, the AVCMX, which represents the crop sensitivity to the temperature, can be determined by the gradient of biomass accumulation (Huang et al., 2022), especially in spring and autumn with greater temperature changes. For maize, the values of VCMX25 and AVCMX have simply followed the previous studies, while BIO2LAI is subject to recalibration, as its necessity of recalibration has been demonstrated by Yu et al. (2022).

Following the establishment of the general photosynthesis rate, we proceed to fine-tune the distribution of carbohydrates among the leaf, stem, and grain compartments, based on the annual cycle of leaf mass and stem data obtained from the station data. Any remaining carbohydrates are allocated to the root. In cases where the recalibration of the distribution scheme alone does not yield satisfactory predictions, adjustments to the turnover and translocation rates are implemented. Additionally, the crop yield will be validated through comparisons with remotely sensed estimations from Grogan et al. (2022).

## 2.4 Modification of the irrigation model

Since our study focuses on the NCP, which predominantly practices dryland cultivation (Zhu et al., 2014), the irrigation methods will mostly pertain to dryland irrigation, excluding grassland irrigation and paddy field irrigation (Huang et al., 2021). To avoid difficulties in modeling canopy interception and surface losses inherent in sprinkler and fast flooding techniques, we opt for drip irrigation using the Noah-MP version 5.0 model (He et al., 2023). This choice simplifies the system while maximizing water resource utilization. The default irrigation module is employed from the planting date to the harvesting date. In order to establish a stronger connection between irrigation and crop growth, irrigation is initiated when the crop is planted and stopped when the crop is harvested. Thus, a reciprocal relationship between crop growth and irrigation is established. As an example, the introduction of irrigation can lead to a cooling effect, consequently decelerating the GDD (Growing Degree Day) accumulation, slowing down crop growth and extending the crop season. This, in turn, requires a longer irrigation period.

The default irrigation can be activated anytime when soil moisture is below a certain threshold within the growing season, which might not be realistic in large-scale applications. In accordance with previous investigations, we add constraints that the irrigation is implemented solely during the local time window of 6 A.M. to 10 A.M. to minimize evaporative losses (Ozdogan et al., 2010; Qian et al., 2013; Yang et al., 2016). Furthermore, the inclusion of winter cultivation necessitates the imposition of temperature limitation, as irrigation under freezing conditions is deemed impractical and detrimental to winter wheat (Yang et al., 2016). To make sure the soil is appropriate for irrigation, we check whether the mean temperature of the uppermost soil layer within the preceding 24-hour period exceeds 5°C. Additionally, we follow the rules from the default irrigation model that the irrigation can be promptly suspended in the presence of precipitation exceeding a threshold rate of 1mm/hr.

Irrigation is required when the soil moisture is lower than the predefined irrigation threshold called management allowable deficit (MAD). MAD is a decimal number between 0 and 1, indicating the cursor between the wilting and the saturated soil moisture. The expected soil moisture after irrigation (SMCLIM) is defined by the MAD, and the soil water deficit is the gap between current soil moisture availability (SMCAVL) and SMCLIM. The total irrigation amount is the integrated deficit of all soil layers. Thus, the default daily irrigation amount is resolved as follows, based on the soil moisture and vegetation fraction which is fixed to be 0.95:

$$\int (\text{SMCLIM} - \text{SMCAVL}) * 0.95$$

When adopting it to large-scale irrigation, we improve the irrigation amount by replacing the constant 0.95 with IRRFRA, i.e., the irrigation land fraction map around year 2005 from the Food Agriculture Organization database (Siebert et al., 2013) as follows

$$\int (\text{SMCLIM} - \text{SMCAVL}) * \text{IRRFRA}$$

It is also stated that the county-level calibrated MAD significantly enhances the irrigation prediction (Xu et al., 2019; Zhang
et al., 2020). Similarly, we calibrated the irrigation threshold province by province using the updated irrigation function, and
finally applied this MAD spatial map to IRRnew. As a comparison, IRRdef only adopts 0.8 as a uniform threshold which is
simply calibrated by the national total amount (Fan et al., 2023).

## 3 Results

### 3.1 Irrigation Simulation

Figure 4 visually illustrates the enhanced predictive capability of our model in accurately capturing the irrigation pattern. It
is challenging to obtain a grid-based observation irrigation map that covers the entirety of eastern China, thus, we mainly
adopt the province-based statistical dataset (National Bureau of Statistics of China, 2005). However, it is only provided as
annual agricultural water usage which not only comprises irrigation but also husbandry, forestry, and fishery consumption
(National Bureau of Statistics of China, 2005). So firstly, agricultural water withdrawal (Fig. 4a) is converted to net
irrigation (Fig. 4b) by multiplying the provincial ratios from Zhu et al. (2012). For better visualization, irrigation is
redistributed to each crop grid based on the irrigation fraction (Fig. 4c). In other words, the weighted provincial mean value
of the redistribution map (Fig. 4d) is the same as the statistical irrigation usage (Fig. 4b). Surprisingly, in Fig. 4d, the annual
irrigation outside the NCP, such as the southern coastal region, is much more intense than that in the NCP region. This is
likely because the statistical "Irrigation Withdrawal" also includes the great consumption used for other crop types such as
raising rice in the extensive paddy field. Our model, however, is currently designed to primarily simulate dryland irrigation
and may not accurately represent water usage in other specific crop types (Yao et al., 2022). Thus, for provinces outside the
NCP, we induce another satellite-based dataset, while keeping the realistic statistic for our targeted NCP region. Its irrigation
amount is grid-based (Fig. 4e) and highly similar to the irrigation land fraction, but it probably has greater uncertainty since
it's not a direct measurement but an empirical estimation based on the water budget orientally (Zhang et al., 2022).
Conclusively, the statistical irrigation in the targeted  NCP (i.e., Beijing, Tianjin, Hebei, Shandong, and Henan, follows Wu
et al., 2018a) is coupled with the satellite-based irrigation in other regions to be the final irrigation map we used for
calibration and validation (Fig. 4f).

The default irrigation scheme (Fig. 4g) exhibits a tendency to overestimate irrigation in the central NCP, deviating from the
observed pattern where irrigation is more prevalent in the western part along the mountain. As expected, the implementation
of the spatially varied irrigation threshold demonstrates a considerable improvement (Fig. 4h), closely resembling the
observed spatial variability. Figure 4i presents the province-based MAD threshold we adopted, which is calibrated using the

observation. Certain provinces in the NCP exhibit higher thresholds, even approaching 1, indicating the model's attempt to achieve near-saturation of the soil. To assess the uncertainty raised from the initial conditions, we conducted nine other

355 simulations starting on consecutive days beginning from March 2nd until March 10th, together with IRRnew starting from March 1st, composing a 10-member ensemble with different initial conditions. The ensemble variability depicted in Fig. 4j is predominantly less than 0.05 mm/day, which is notably smaller in comparison to the annual irrigation amount. This suggests that the spin-up time utilized in the simulation is sufficient, and the initial conditions do not introduce significant uncertainty to the irrigation simulation, which further reinforces the reliability and robustness of the model in capturing the irrigation

360 dynamics.

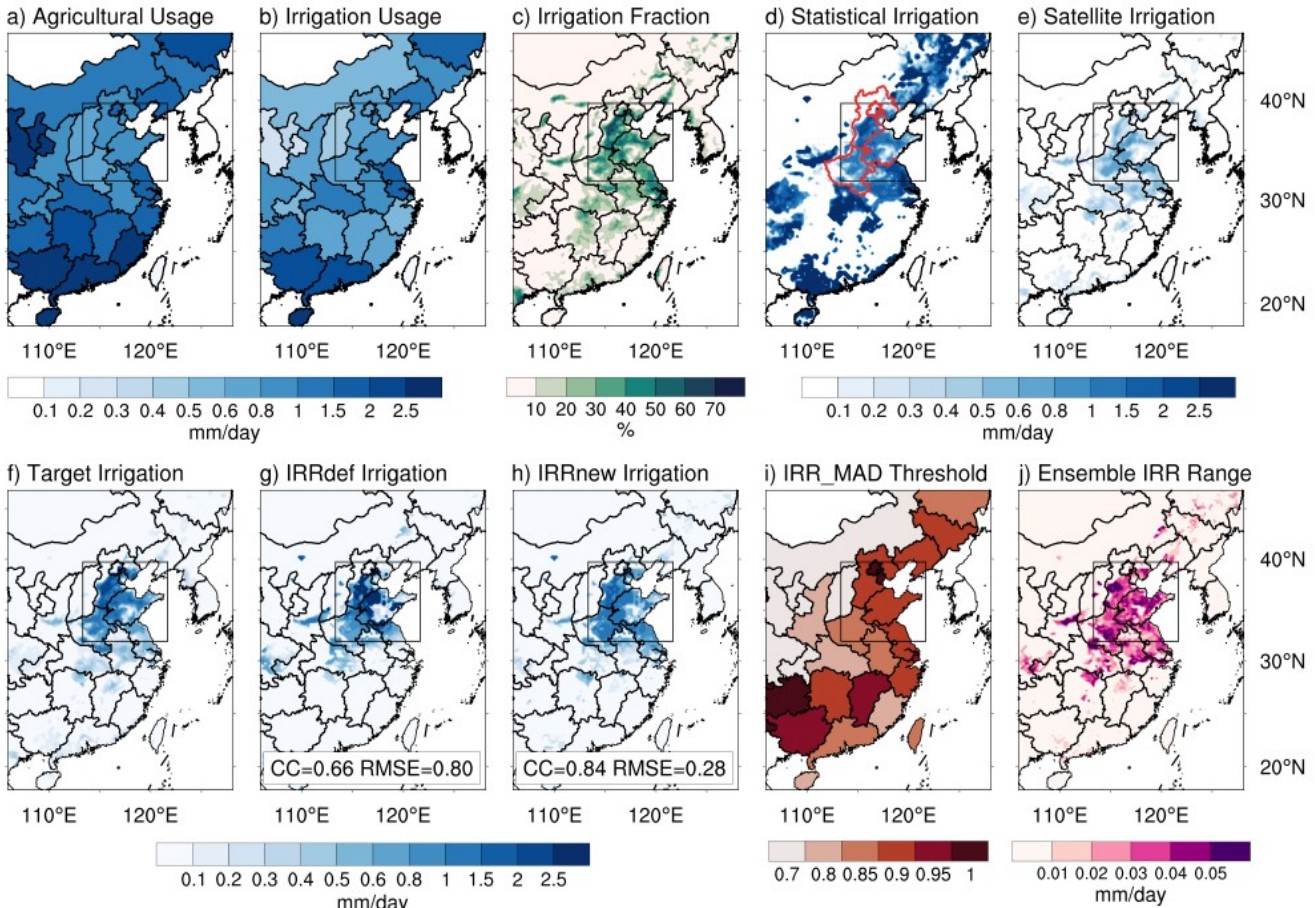

**Figure 4. Spatial maps of 2005 (a) agricultural usage, (b) estimated irrigation usage, (c) irrigation fraction (same as Fig. 1d), (d) statistical irrigation, e) satellite irrigation, f) target irrigation, (g) simulated irrigation using the default irrigation scheme (IRRdef), (h) simulated irrigation using improved irrigation scheme (IRRnew), (i) MAD (Manageable allowable deficit) irrigation**

365 **threshold adopted in IRRnew and (j) irrigation range among 10 ensemble members using different initial conditions. For easy comparison, all subplots with blue colors (Fig. 4a,b,d,e,f,g,h) adopt the same color scale.**

Figure 5 offers a visual representation of the long-term impact of the scheme improvement on the irrigation pattern, showcasing the average results over a 10-year period. The lines depict the monthly irrigation levels, while the bars represent the averaged LAI across all crop grids in the NCP region. The default irrigation scheme tends to apply excessive irrigation during the winter season, which can be attributed to the relatively drier soil conditions and thus larger gap between the soil moisture and the MAD threshold. However, irrigation under freezing conditions is deemed impractical and detrimental to winter wheat (Yang et al., 2016). Thus, despite the intense winter irrigation, the corresponding vegetation growth, as indicated by the LAI, shows insignificant improvement in winter. On the other hand, the improved model effectively avoids unnecessary winter irrigation, allowing for a greater allocation of water resources during the spring, summer, and fall seasons when crop growth is more pronounced. Consequently, this strategic water distribution leads to more flourishing vegetation, especially during the summer cropping season. In summary, the improved model provides enhanced water support to the crops while also conserving irrigation consumption on an annual basis.

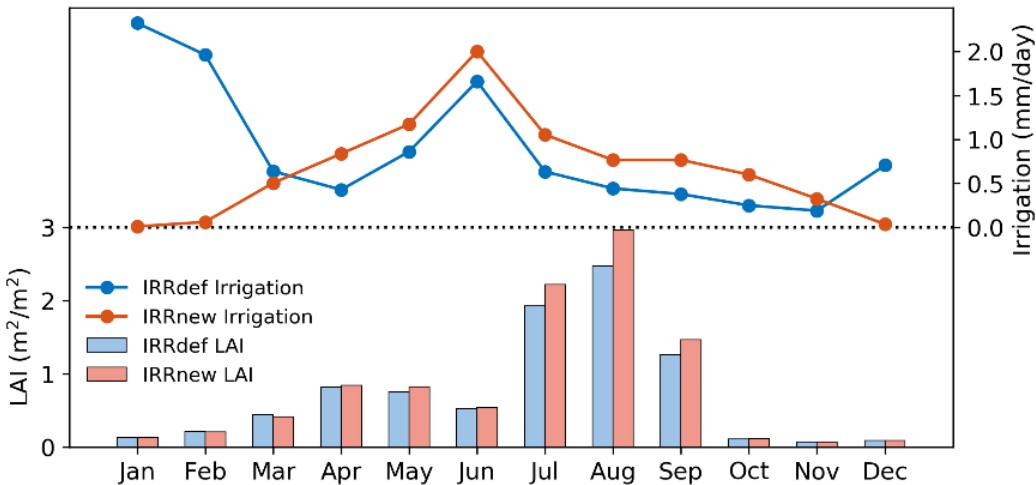

**Figure 5. Monthly irrigation (lines) and LAI (bars) using default irrigation scheme (IRRdef) and improved irrigation scheme (IRRnew). Monthly values are the average of all crop grids in the NCP over the period of 2005-2014.**

Figure 6 presents irrigation impact on energy partition by depicting the differences between the irrigation simulation (IRRnew) and the non-irrigation simulation (CROPnew). The upper panel visualizes the spatial changes, while the lower panel illustrates the monthly averaged changes for the entire NCP region (represented by the blue line) and the double-cropping region (represented by the orange line). As expected, the increased soil moisture contributes to a higher latent heat flux, with a maximum increase over 40 W/m². Conversely, irrigation-induced evaporation cools the surface, leading to a reduction in sensible heat flux, with the sharpest decrease around 30 W/m². The cooler surface also reduced longwave radiation emitted from the surface, causing increases in net radiation with the greatest change about 15 W/m². Overall, the increase in latent heat flux surpasses the decrease in sensible heat flux, and when combined, their changes partially balance

out to equal the net radiation. The most substantial changes are observed in southern Hebei province, which aligns with the irrigation fraction map (Fig. 4c). In the lower panel, all monthly patterns exhibit two peaks, with a larger peak in June and a smaller peak in September. The monthly pattern within the double-cropping area shows more pronounced changes and a more distinct two-peak structure. Furthermore, the irrigation responses of all variables display similar spatial and temporal patterns to the irrigation amount, indicating a strong correlation between irrigation application and these observed changes.

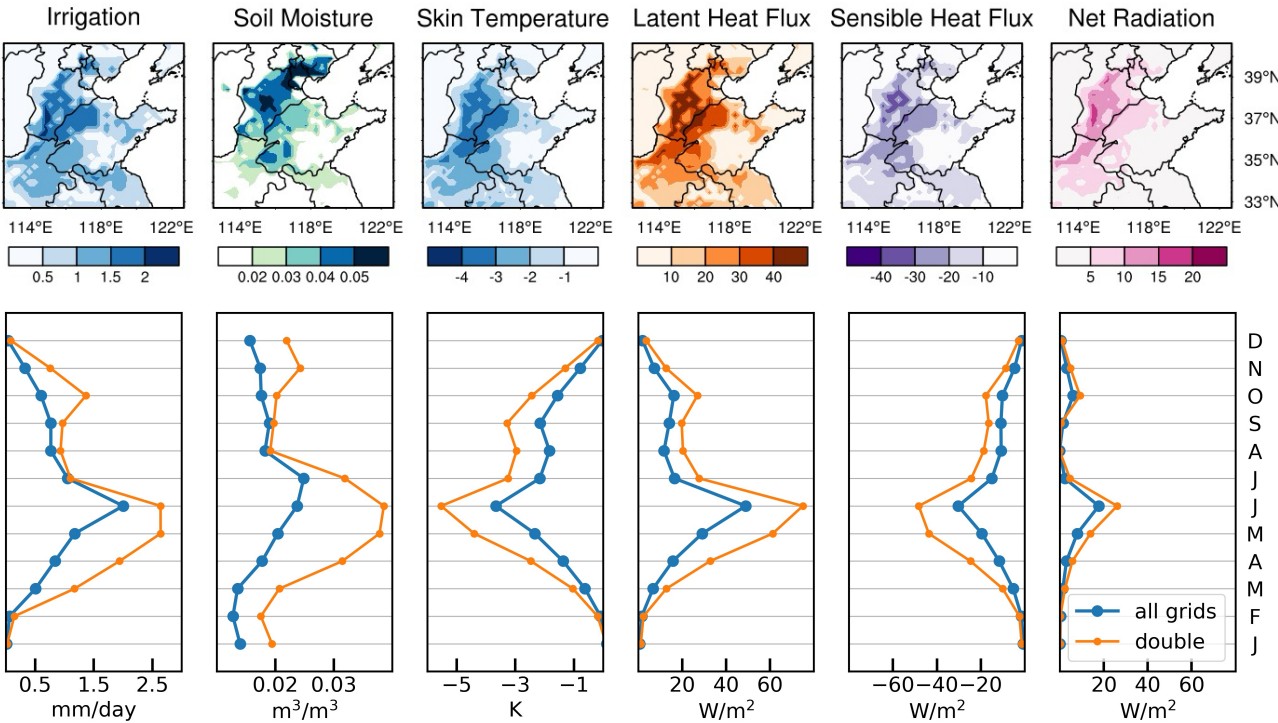

**Figure 6. Irrigation-induced changes (IRRnew-CROPnew) in the climatology spatial pattern (upper panel) and mean monthly pattern (lower panel) of various variables, including irrigation, soil moisture, skin temperature, latent heat flux, sensible heat flux, and net radiation. The blue line represents the average value for all grids in the North China Plain (NCP), while the orange lines correspond to the double-cropping area only.**

## 3.2 Evaluation of crop growth

The evaluation of the crop simulation encompasses several key aspects, including crop stage identification, annual cycle of leaf and stem mass, crop yield prediction, and general LAI simulation. These components will be scrutinized to assess the validity and accuracy of the crop simulation.

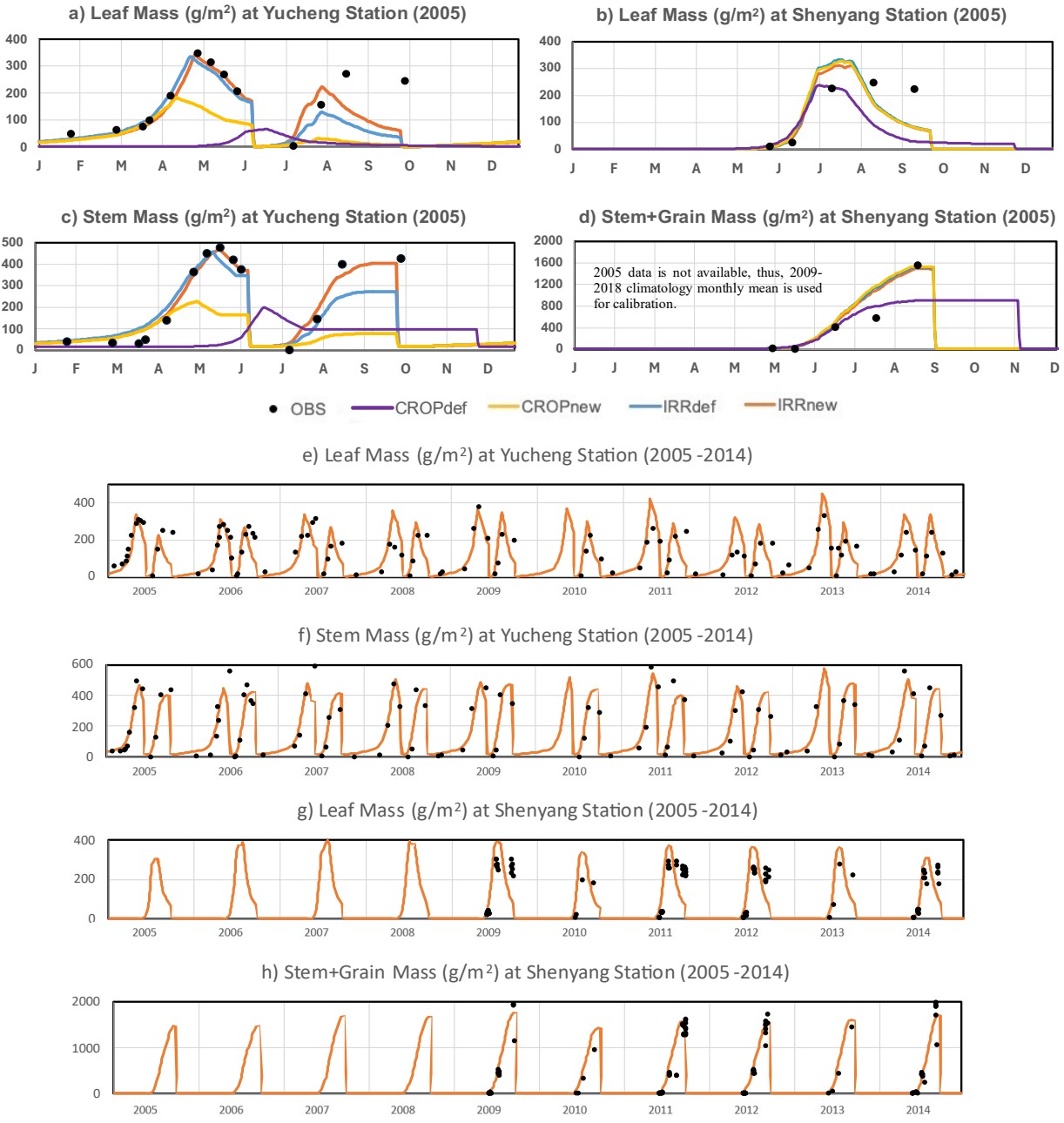

**Figure 7. Biomass comparison between simulation and station data at Yucheng and Shenyang. Black dots indicate station observations, while the lines represent the simulation results. Panels (a-d) illustrate the annual cycle of each simulation for the year 2005 as well as the corresponding station data. Panels (e-h) present the ten-year biomass of the IRRnew (with improved crop and improved irrigation model) alongside the station data.**

### 3.2.1 Validation of Biomass in Yucheng and Shenyang

The station-based biomass in year 2005 is adopted for calibration (Fig. 7a-d). The biomass cycle in Yucheng clearly exhibits two distinct peaks, representing two crop seasons. Implementation of double-cropping function reshapes the pattern from single-peak to double-peak, and the application of irrigation extends the winter wheat growth, shifting the peak to the right side and resulting in a better match with the observation. Furthermore, the improvements in the irrigation model lead to significant enhancements at the Yucheng, particularly for summer maize. This aligns with the previous conclusions, as well

as the suboptimal maize growth under water stress conditions captured by another crop model (Song and Jin, 2020), further approving the positive influence of the improved irrigation model on crop growth. On the other hand, irrigation is not intensely adopted in northeast China, and thus, does not make a noticeable impact in Shenyang (Figs. 7b and 7d). The long-term biomass results, displayed in Figs. 7e-g, provide additional long-term validation for the crop simulation. While the model does not fully capture the inter-annual variability, it does exhibit some fluctuations that align with observed patterns.

For instance, the winter wheat crop in Yucheng shows poorer growth in 2012, while the crop in Shenyang performs worse in 2010.

### 3.2.2 Validation of crop calendar and grain mass

To evaluate the performance of the stage identification process within the crop model, we compare the 10-year mean heading

and maturity dates from each simulation with the satellite estimations (top and middle panels in Fig. 8). Since the model accumulates carbon to grain starting from the initial reproductive stage, we regarded the start of the initial reproductive stage as the heading date, which aligns with the heading date identified by the time of maximum LAI in the satellite estimation. Similarly, the transition day from the post-reproductive stage to the maturity stage is regarded as the maturity date. According to the algorithm, the heading and maturity dates can be regarded as rough indicators of the transition from the

vegetative stage to the reproductive stage, and ultimately to the maturity stage. This validation process allows us to assess the model's ability to accurately simulate the temporal development of crop growth.

Figure 8 shows the progressive improvements made by each step of the model modification in predicting the crop phenology. Typically, winter wheat heads in March and matures in May, while maize heads in August and matures in

September. The default crop model only considers single cropping without winter wheat. Moreover, the heading date of CROPdef is observed to be one or two months earlier than the observations, and the maturity date also exhibits deviations, being earlier in the NCP. This suggests that employing a uniform starting and ending time is not suitable for a regional domain. The enhanced crop model, CROPnew, incorporates double cropping and spatially varied planting and harvesting dates, resulting in more accurate crop growth duration across the two seasons. The early bias is further mitigated by

irrigation, as the presence of moist soil induces primary cooling, subsequently decelerating temperature accumulation and postponing the growth stage.

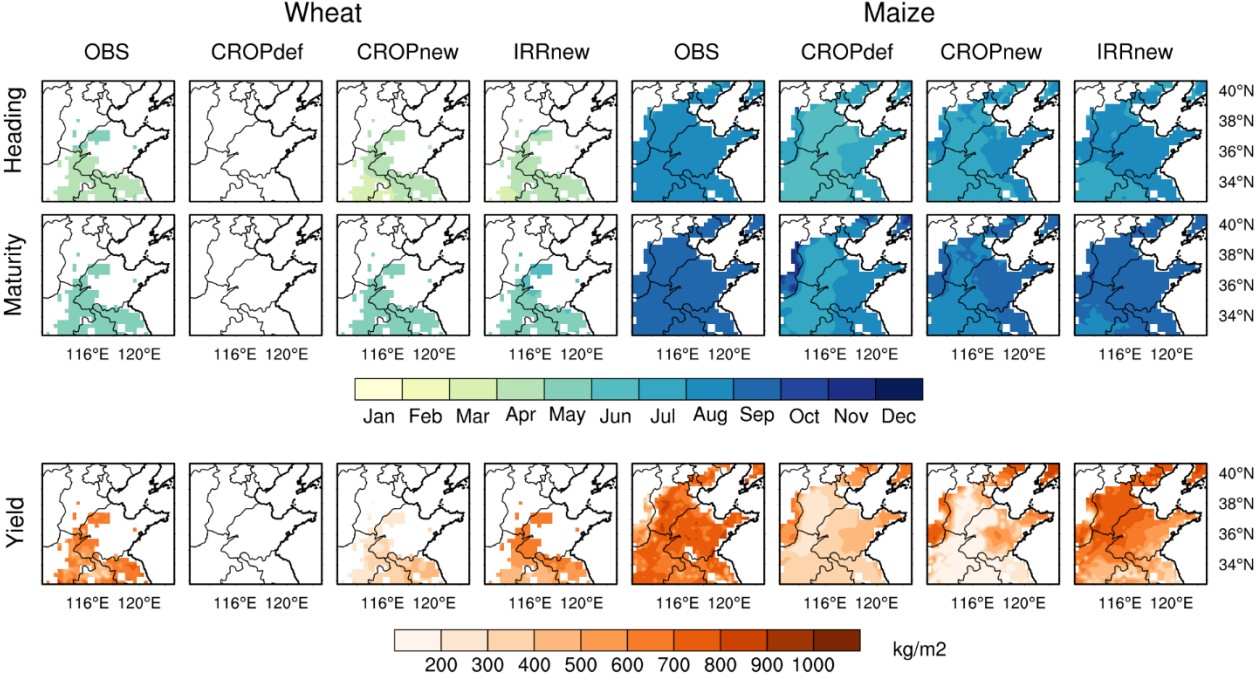

**Figure 8. Comparison of the crop growth calendar and yield by comparing the heading date, maturity date, and annual yield for wheat and maize between observation (OBS) and simulation (CROPdef using default crop model, CROPnew using improved crop model, and IRRnew using both improved crop model and improved irrigation model).**


Similar enhancement can be observed when assessing the crop yield (bottom panels in Fig. 8). Due to the limited availability of grid-scale yield data, the computed 2015 crop yield from the Global Agro-Ecological Zones (GAEZ) model is used as the observational benchmark. The initial CROPdef only considers a single maize season, and it proves to be inadequate for the heavily irrigated NCP region, even with the exaggerated assumption of a fixed FVEG value of 0.95. Despite the recalibration

of parameters and adjustments to the planting and harvesting dates, which realizes the double cropping simulations in the CROPnew, production in the NCP region is still severely hindered by the limited water availability. Similar to the previous validation of crop calendar, the activation of the irrigation in IRRnew noticeably promotes the crop growth. This highlights the importance of irrigation in sustaining the compact rotation and high productivity in the NCP. In conclusion, each of the following factors—implementation of double cropping, adoption of spatially varying input, and integration of irrigation—

holds significant importance in accurately simulating the crop calendar and grain yield.

### 3.2.3 Validation of long-term LAI and FVEG

In comparison to winter wheat, the simulation of maize does not exhibit a perfect match with the observed data, as fewer parameters have undergone recalibration. However, despite these imperfections, the model demonstrates a reasonable performance in simulating crop growth, especially when considering its overall predictability across the entire NCP region. This is evident in the validation of monthly LAI whose accuracy plays a crucial role in determining land-atmosphere interaction and energy partitioning (Liu et al., 2016). Figure 9 compares the simplest crop model and the final integrated system with observation, emphasizing the remarkable improvement achieved through the integration and regionalization processes. Figure S1 provides an extended version inclusive of all simulations and whole simulation domain, thoroughly visualizing the gradual improvement made by each step. The observed LAI demonstrates a gradual increase until May, with a slight decline in June, indicating the harvest of winter wheat. In the second crop season, there is a notable rise in LAI during July and August, reflecting substantial growth and vegetation development during this period, followed by a gradual decline in September and October.

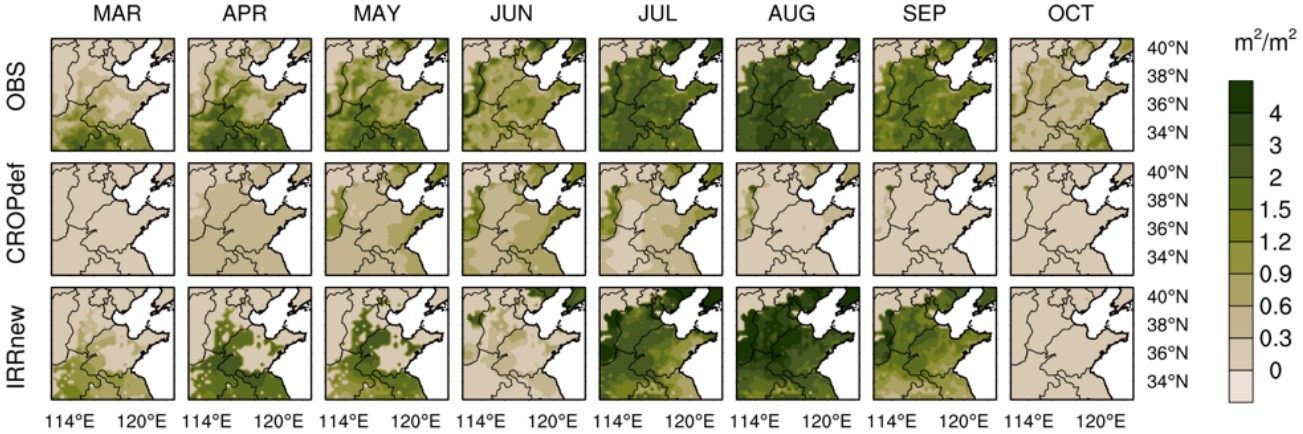

**Figure 9. Monthly LAI patterns of the satellite observation (OBS), simulation with default crop model only (CROPdef), and simulation with improved crop and improved irrigation (IRRnew) from March (MAR) to October (OCT).**

It becomes evident that the CROPdef lacks representation of the first crop season and exhibits an early and truncated second crop season in the NCP. The inclusion of irrigation, both in the IRRdef and IRRnew models, significantly enhances crop growth in the double cropping region, highlighting the crucial role of irrigation in this region. Conversely, the crops in Northeast China, where rain-fed agriculture predominates, exhibit reasonably satisfactory growth even without irrigation. This regional disparity in crop sensitivity to irrigation can be aptly captured by the improved system. In line with the previous figures, the IRRnew proves particularly beneficial for the growth of summer maize. Its avoidance of unnecessary irrigation during the freezing winter months allows for greater resource allocation during the productive summer period, resulting in improved growth and development. Generally, the IRRnew simulation successfully captures the spatial and

temporal LAI patterns, particularly in the NCP region, which demonstrates a superior capability in accurately representing the dynamics of crop growth compared to the initial crop model. In addition to the LAI, the joint crop modeling system also demonstrates reasonable predictability in monthly FVEG (Fig. S2). Consequently, this expanded functionality offers valuable opportunities to conduct sensitivity tests, enabling a deeper understanding of the agriculture-related climate response.


### 3.2.4 Quantitative validation of long-term irrigation and yield

To further quantify the accuracy and stability of the simulation, Fig. 10 compared the irrigation intensity and crop yield from the IRRnew results with the province-based statistics data spanning the entire period from 2005 to 2014. Each dot represents one province and most provinces are simply depicted gray dots. Three provinces with large cropland extent in the NCP—
Shandong, Henan and Hebei—are depicted in red dots with horizontal and vertical error bars showing the inter-annual variability of observation and simulation, respectively. Most of the dots, especially the red dots, are located in close proximity to the diagonal line, indicating a reasonably accurate predictability of irrigation amounts and crop yields. The comparable lengths of the horizontal and vertical error bars suggest that the uncertainties associated with the observation and simulation, respectively, are at least comparable. Furthermore, the model demonstrates greater accuracy and reliability in
simulating winter wheat, which underwent more comprehensive calibration, compared to the maize.

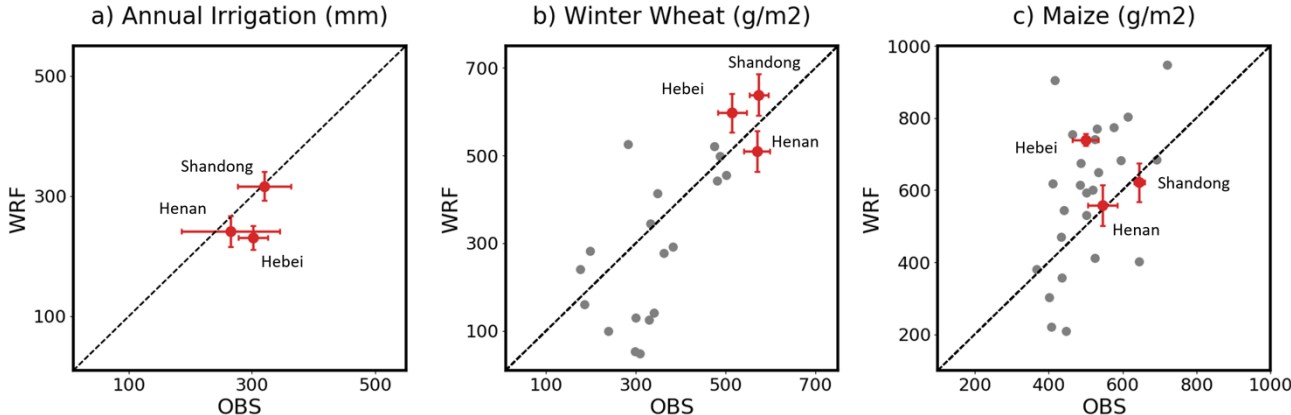

**Figure 10. Validation of the climatological mean of annual irrigation and crop yield across provinces. The red dots correspond to the three provinces with extensive cropland coverage in the North China Plain (NCP), while the horizontal and vertical error bars depict the inter-annual variability of observation and simulation, respectively. The gray dots represent the remaining provinces.**

## 4 Discussion and conclusion

The validation process has brought to light several limitations of the current model. To start with, the model design restricts the simulation to only one crop type per grid. This simplification may contribute to inaccuracies in predicting the leaf mass of summer maize at the Yucheng, which can be revealed by the inconsistency of LAI observation (Fig. 9) in the NCP region and the leaf mass at the Yucheng (Fig. 7). While the LAI values indicate that September should have a smaller LAI compared to July (Fig. 9), the station data suggests that September actually has a greater leaf mass than July (Fig. 7). This discrepancy is likely attributed to two factors. Firstly, the specific leaf area, or BIO2LAI in the model, varies across different crop stages, as supported by both station data and existing literature (Amanullah, 2015; Zhou et al., 2020). In other words, the leaves may be thinner in July, while they become thicker in September. The second reason is that the observed LAI pattern represents a spatial average value over the grid, which may contain a diverse range of crops. Consequently, the specific station data for summer maize may not align well with the spatially averaged LAI. Since this study primarily focuses on the regional scale rather than individual field points, we prioritize matching the spatial LAI pattern while partially sacrificing the accuracy in predicting station biomass. As a result, the simulated LAI pattern is well-matched in the NCP region, while the predicted leaf mass for summer maize may not closely align with the station data. On the contrary, winter wheat greatly, even exclusively dominates the first crop season, and thus the station data and spatial pattern are consistent and can both be captured by the model (Fig. 7 and Fig. 9). Also, the predicted LAI is completely cleared up after harvesting, since each grid can only predict one type of growth pattern, which is different from the gradual fading observed in June and October.

It is important to acknowledge that the model performance may be less satisfactory in southern NCP. There is some underestimation of LAI compared with northern China. This could potentially be attributed to the limited predictability of FVEG. Even in regions where the model currently exhibits reasonable performance, uncertainty can arise from the model's sensitivity to soil moisture (Wang, 2005). Adopting satellite-based estimated irrigation datasets may also introduce uncertainty, thus, it becomes crucial to conduct model sensitivity tests under varying water forcings for future irrigation impact studies. To enhance our understanding of the irrigation impact on regional climate, our study focuses on simulating irrigated crop growth in the NCP region using the WRF model. In order to improve the model's capabilities, we have implemented the following enhancements:

- Incorporating the winter crop season and facilitating double cropping, which was previously absent in the WRF system.
- Establishing a linkage between the FVEG and crop-based LAI to capture spatial and seasonal variations, as well as enable its sensitivity to water forcings.
- Calibrating parameters and utilizing local input data for winter wheat and maize to accurately represent the general vegetation patterns in the NCP region.

- Integrating the irrigation scheme with the crop simulation, activating irrigation based on the crop stage to account for the climate's impact on the irrigation season.
- Implementing a temperature check before irrigation to prevent harmful irrigation during freezing periods.
- Calibrating the irrigation threshold on a province-by-province basis to ensure more realistic estimates of irrigation amounts.

These enhancements significantly improve the model's performance in identifying crop stages, estimating field biomass, predicting crop yield, and projecting monthly leaf area index. Importantly, our study demonstrates the reasonable
performance of this regional-scale application in the NCP region, despite the distinct climate background compared to the model's original development in the central US. This implies the potential application of the WRF in other agricultural zones. And most of our validation data is derived from satellite observations, indicating the possibility of adopting this model in regions even with limited ground-based data. Also, the integrated crop system clearly highlights the significance of an appropriate irrigation scheme in the NCP region. Future studies will connect the irrigated system with the groundwater layer,
since the NCP heavily relied on groundwater-supplied irrigation. Groundwater depletion can also lead to hydrological changes (An et al., 2021; Famiglietti, 2014), further impacting the interactions between cultivation and climate.

**Acknowledgments**

This study was supported by the Hong Kong Research Grants Council funded project, GRF16309719. Additionally, E.-S. Im and J. Hur were partly supported by the "Research Program for Agricultural Science & Technology Development (Project
No. PJ014882)", National Institute of Agricultural Sciences, Rural Development Administration, Republic of Korea. Z. Y. was supported by the Office of Science of the U.S. Department of Energy (DOE) as part of the Atmospheric System Research (ASR) Program via Grant KP1701000/57131. We would like to give special thanks to Dr. Ben Yang for providing his irrigation model as a reference.

**Data Availability Statement**

The climatology precipitation is retrieved from the China Meteorological Forcing Dataset and is adopted for precipitation validation. It is produced by Cold and Arid Regions Science Data Center, with doi:10.3972/westdc.002.2014.db, published at http://westdc.westgis.ac.cn. East Asian Summer Monsoon Index is referred to http://lijianping.cn/dct/page/65577, with the definition from Li and Zeng (Li and Zeng, 2002). LAI dataset is initially Sun Yat-sen University (Yuan et al., 2020), shown at http://globalchange.bnu.edu.cn/data/global_lai_0.1/. The cropping pattern is defined by ChinaCP (Qiu et al., 2022),
available at https://doi.org/10.6084/m9.figshare.14936052. The planting and harvesting date is from the ChinaCropPhen1km dataset (Luo et al., 2020) at https://doi.org/10.6084/m9.figshare.8313530). Station data is provided by the National

Ecosystem Science Data Center, National Science & Technology Infrastructure of China. ( http://www.nesdc.org.cn),  And the yield data (Grogan et al., 2022) is freely available from https://doi.org/10.7910/DVN/XGGJAV.

**Code Availability Statement**

The source code for double cropping with interactive irrigation is published on https://doi.org/10.5281/zenodo.10729554.

**Author contribution**

YF and EI conceptualize the idea; YF developed the code, performed the experiment and wrote the manuscript draft; all authors reviewed and edited the manuscript.

**Competing interests**

Some authors are members of the editorial board of journal *Geoscientific Model Development*.

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
