# Peer review of "Applying double-cropping and interactive irrigation in the North China Plain using WRF4.5"

_Geoscientific Model Development, 2024_

## Referee Comment (RC2)

Comments on Manuscript GMD-2024-38
**"Applying double-cropping and interactive irrigation in the North China Plain using WRF4.5"**
by Yuwen Fan, Zhao Yang, Min-Hui Lo, Jina Hur, and
Eun-Soon Im

**Recommendation**: Accept with minor revision.

This study describes the usefulness of the WRF-Crop model in capturing vegetation and irrigation patterns in the North China Plain (NCP), by incorporating double-cropping with interactive irrigation. The authors modified the crop model in terms of the vegetation fraction (FVEG) and the planting/harvesting dates and improved the irrigation model in calculating the irrigation amount. The authors validated their model results in terms of various irrigation and crop growth aspects and concluded that coupling of the enhanced crop and irrigation models significantly improved the performance in estimating crop stages and yields, field biomass, and leaf area index. This manuscript can be a valuable report to the scientific community for better prediction of double cropping and irrigation aspects in NCP; however, some issues need to be clarified or discussed in more detail.

**Specific comments:**

1. The authors need to specifiy major differences in their methods and results compared with those of Yu et al. (2022).

2. Abstract: Delete the last sentence describing the future research. Just include more details and focus on the current research.

3. Plain Language Summary: This part should have more scientific information, including more details in results and their implications.

4. L 53 & L 78–79: Remove the commas in front of 'but alo'. Note that 'not only … but also' requires a comma only when two independent clauses are linked. No comma is required in linking nouns or noun phrases.

5. L55: "while others incorporate irrigation with fixed amount (Vira et al., 2019) or dynamically based on daily soil conditions" ⟶ Hard to understand: Rewrite. Should 'dynamically' be replaced with 'dynamically varying amount' or something else?

6. L 69: "regionalizing the algorithms" ⟶ How can the algorithms be regionalized? In general, one develops new parameterization schemes or improves the existing parameterization schemes and/or tunes/calibrates/optimizes the parameter values in the schemes, making the schemes and/or parameter values work well in a specific region of interest. The authors need to explain more explicityly on 'regionalization of algorithms'.

7. L 71: Include a paragraph that provides examples of 'regionalizing algorithms', e.g., developing new algorithms for a specific region and tuning/calibrating/optimizing parameter values that fits observations well in a specific region. Some examples of developing parameterizatin schemes at regional scales, in chronical order, include

Bou-Zeid, E., Parlange, M. B., and Meneveau, C.: On the Parameterization of Surface Roughness at Regional Scales. J. Atmos. Sci., 64, 216–227, https://doi.org/10.1175/JAS3826.1, 2007.

Liu, J., Ding, Y., Zhou, X., Li, Y.: A Parameterization Scheme for Regional Average Runoff over Heterogeneous Land Surface Under Climatic Rainfall Forcing, J. Meteorol. Res., 24, 116-122, 2010.

Song, W., Tang, H., Sun, X., Xiang, Y., Ma, X., Zhang, H.: Developing a New ParameterizationScheme of Temperature Lapse Ratefor the Hydrological Simulation ina Glacierized Basin Based on Remote Sensing. Remote Sens., 14, 4973, `https://doi.org/10.3390/rs14194973`, 2022.

Asmus, C., Hoffmann, P., Pietikäinen, J.-P., Böhner, J., and Rechid, D.: Modeling and evaluating the effects of irrigation on land–atmosphere interaction in southwestern Europe with the regional climate model REMO2020–iMOVE using a newly developed parameterization, Geosci. Model Dev., 16, 7311–7337, https://doi.org/10.5194/gmd-16-7311-2023, 2023.

and some examples of regional parameter estimations, in chronical order, include

Xie, Z., Yuan, F., Duan, Q., Zheng, J., Liang, M., and Chen, F.: Regional Parameter Estimation of the VIC Land Surface Model: Methodology and Application to River Basins in China. J. Hydrometeor., 8, 447–468, https://doi.org/10.1175/JHM568.1, 2007.

Livneh, B., and Lettenmaier, D. P.: Regional parameter estimation for the unified land model, Water Resour. Res., 49, `https://doi.org/10.1029/2012WR012220`, 2013.

Hong, S., Park, S. K., and Yu, X.: Scheme-Based Optimization of Land Surface Model Using a Micro-Genetic Algorithm: Assessment of Its Performance and Usability for Regional Applications, SOLA, 11, 129-133, https://doi.org/10.2151/sola.2015-030, 2015.

Park, S. and Park, S. K.: A micro-genetic algorithm (GA v1.7.1a) for combinatorial optimization of physics parameterizations in the Weather Research and Forecasting model (v4.0.3) for quantitative precipitation forecast in Korea, Geosci. Model Dev., 14, 6241–6255, https://doi.org/10.5194/gmd-14-6241-2021, 2021.

8. L 84: The acronym 'LSM' should be replaced with 'LSMs'.

9. L 89–90: "Noah-Multiparameterization (Noah-MP)" ⟶ Give the full original words "Noah land surface model with multiparameterization options (Noah-MP)", and provide a proper reference for it, i.e., Niu et al. (2011).

10. L 92–93: "Weather Research Forecast (WRF)" ⟹ "Weather Research and Forecasting (WRF) model", i.e., provide the correct original words for WRF. And cite a proper reference.

11. Figure 1: Modify the caption to "(a) Annual precipitation (mm/day) and basic geostatic variables applied in this project, including (b) topography (m), (c) cropland fraction (%), and (d) irrigated land fraction (%)."

12. L 120: Provide the proper reference of WRF version 4.3. Please also check if this version number is correct because your title says it is WRF4.5.

13. L 127: "ERA5-Interim" ⟶ There is no ERA5-Interim reanalysis data. It should be either "ERA5" or "ERA-Interim". Based on the cited reference, Hersbach et al. (2020), it should be "ERA5".

14. L 143: There exists only one Liu et al. (2016) and only one Zhang et al. (2020) in the References; thus, the authors do not need to put the first names' initials. Replace "X. Liu et al. (2016)" with "Liu et al. (2016)" and "Z. Zhang et al. (2020)" with "Zhang et al. (2020)". Please do it throughout the manuscript.

15. L 175: There exists only one Wu et al. (2018b) in the References; thus, replace "L. Wu et al. (2018b)" with "Wu et al. (2018b)".

16. L 194: "crop growth." $\Longrightarrow$ "crop growth. Equations (1) and (2) below represent the original FVEG equation by Niu et al. (2011) and the adjusted FVEG suggested in this study, respectively:"

17. L 196, Eq. (1): "Niu et al. (2011) FVEG" $\Longrightarrow$ "Original FVEG"

18. L 222–223: Describe explicity 'Yucheng' and 'Shenyang' in Fig. 3a. In Fig. 3a, two red circles may represent these two locations, but the authors needs to put the location names in the map. Most readers around the world are not familiar with the location names.

19. L 225: The double-cropping area is quite large: Would Yucheng be considered to represent well the large area, especially the southern part of the area? Isn't there any station near the southern edge of the area? If another station exists with long-term observation at the southern part of the area, including it will make this study more valuable. Otherwise, add a sentence that Yucheng, located at the northern part of the double-cropping area, can well represent the characteristics of the southern part of the area, with some scientific evidences.

20. L 231: The authors used 'Vcmx25' and 'BIO2LAI' without any definition or explanation. If they are acronyms, please provide the original worrds; otherwise, explain what they are.

21. L 237: "Table S1 provides ... with the supporting scientific references and recalibration procedures." $\Longrightarrow$ "Table S1 provides ... with the supporting scientific references." Then, from Supplementary, please move the 2 paragraphs below "**The recalibration process:**" to the end of Section 2.3.3.

22. Figure 3: In the caption of Fig. 3, explain what the two red circles in Fig. 3a means.

23. Figure 3: Figures 3b-e are never cited in the manucript. The authors need to properly cite each subfigure in the text, probably in the paragraph in L 210–219.

24. L 249: Use consistent tense: "the crop emerges" vs. "the crop matured".

25. L 254: Is there any reason that 'irrigation' should be expressed in capital ("The default Irrigation")?

26. L 267: What does the number '2005' mean? Is it the year 2005?

27. Equations 3 & 4: Define $SMCLIM$ and $SMCAVL$ immediately following the equations.

28. L 264–269: It is recommended to modify this part as below:

   The default daily irrigation amount is resolved as follows, based on the soil moisture and vegetation fraction which is fixed to be 0.95:

   $$\int (SMCLIM - SMCAVL) * 0.95 \qquad (3)$$

   where $SMCLIM$ is $\cdots$ and $SMCAVL$ is $\cdots$. When adopting it to large-scale irrigation, we improve the irrigation amount by replacing the constant 0.95 with $IRRFRA$, i.e., the irrigation land fraction map around 2005 from the Food Agriculture Organization database (Siebert et al., 2013) as follows:

   $$\int (SMCLIM - SMCAVL) * IRRFRA. \qquad (4)$$

29. L 272–273: Definition of $SMCLIM$ and $SMCAVL$ should appear immediately following Equations 3 & 4.

30. L 284: "which not only comprises irrigation, but also husbandry, forestry, and fishery consumption" $\implies$ "which comprises not only irrigation but also husbandry, forestry, and fishery consumption"

31. L 285–287: Figures 4b and 4c are cited before Fig. 4a is cited. It is recommended to switch the order of subfigures in the order that they appear in the explanation.

32. L 296–297: "NCP (i.e., Beijing, Tianjin, Hebei, Shandong, and Henan, follows D. Wu et al., 2018) is coupled" $\implies$ "NCP—Beijing, Tianjin, Hebei, Shandong and Henan that follow Wu et al. (2018)—is coupled"

33. L 307: "Figure 4(j)" $\implies$ "Figure 4j"

34. L 307–315: Figure 4j is cited before Figure 4i is cited: Switch the order of these two subfigures in Fig. 4.

35. Figure 4, Caption: "(IRRnew), and (i) irrigation range among 10 ensemble members using different initial conditions (j) MAD" $\implies$ "(IRRnew), (i) irrigation range among 10 ensemble members using different initial conditions, and (j) MAD"

36. Figure 5 and L 319–329: The authors just compared the results between two models—IRRdef vs. IRRnew. To verify that IRRnew is better, the authos should also show the observations. Although the authors showed that IRRnew had better results than IRRdef in terms of spatial distributions in Fig. 4, the authors should also validate the model results in terms of temporal variations.

37. L 337: "(first two lines in Fig. 6)" $\implies$ "(top and middle panels in Fig. 6)"

38. L 338: "we considered the entering the initial reproductive stage as the heading date" $\implies$ "we regarded the start of the initial reproductive stage as the heading date"

39. L 340: "is considered as the maturity date" $\implies$ "is considered the maturity date" or "is regarded as the maturity date"

40. L 341: "can be considered as rough indicators" $\implies$ "can be considered rough indicators" or "can be regarded as rough indicators"

41. L 351: "time Is not" $\implies$ "time is not"

42. L 356: "(third row in Fig. 6)" $\implies$ "(bottom panels in Fig. 6)"

43. L 365: "each of the following factors, implementation $\cdots$ irrigation, holds" $\implies$ "each of the following factors—implementation $\cdots$ irrigation—holds"

44. L 368: "Yucheng and Shenyang station" $\Longrightarrow$ "Yucheng and Shenyang" (We already know that Yucheng and Shenyang are stations.)

45. L 369: "The station-based biomass is adopted for calibration (Fig. 7a-d)." $\Longrightarrow$ "The station-based biomass in a specific year is adopted for calibration in both stations (Figs. 7a-d)."

46. L 369 & 373: "Yucheng station" $\Longrightarrow$ "Yucheng"

47. L 376: "Shenyang Station" $\Longrightarrow$ "Shenyang"

48. L 376: "(Fig. 7b and 7d)" $\Longrightarrow$ "(Figs. 7b and 7d)"

49. L 377: "Fig. 7e-g" $\Longrightarrow$ "Figs. 7e-g"

50. Figure S1: The bottom panels of Fig. S1 show "GWBnew", which is never defined either in the main text or in the Supplementary. Define "GWBnew".

51. Figure 8: The caption can be better described as follows: "Monthly LAI patterns of the satellite observation (OBS), simulation with default crop model only (CROPdef), and simulation with improved crop and improved irrigation (IRRnew) from March (MAR) to October (OCT)."

52. L 417–418: "in the NCP, Shandong, Henan and Hebei, are depicted" $\Longrightarrow$ "in the NCP—Shandong, Henan and Hebei—are depicted"

53. L 418–419: "with horizontal and vertical error bars showing the inter-annual variability of both observation and simulation" $\Longrightarrow$ "with horizontal and vertical error bars showing the inter-annual variability of observation and simulation, respectively" (Please check if this modification gives correct interpretation.)

54. L 419: "Most of the dots especially the red dots, are" $\Longrightarrow$ "Most of the dots, especially the red dots, are"

55. L 421: "the uncertainties associated with the observation and simulation are" $\Longrightarrow$ "the uncertainties associated with the observation and simulation, respectively, are"

56. Figure 9, Caption: "the horizontal and vertical error bars depict the inter-annual variability observed in both the simulations and actual measurements" ⟹ "the horizontal and vertical error bars depict the inter-annual variability observed in the simulations and the actual measurements, respectively"

57. L 430–431: "the model design restricts the simulation of only one crop type per grid" ⟹ "the model design restricts the simulation to only one crop type per grid"

58. L 432 & 433: "the Yucheng Station" ⟹ "Yucheng"

59. L 447: "It is important to acknowledge that the model performance may be less satisfactory in southern NCP." ⟶ This may be because the authors did not include an observation station in the southern part of NCP for calibration (see the comment in #19). It will be great if the authors can include one station in the sourthern NCP for calibration and compare the results.

60. L 453–454: This part should not be itemized, i.e., it should be rewritten as

    > To enhance our understanding of the irrigation impact on regional climate, our study focuses on simulating irrigated crop growth in the NCP region using the WRF-Crop model. In order to improve the model's capabilities, we have implemented the following enhancements:
    >   - Incorporating $\cdots$
    >   - Establishing $\cdots$
    >     $\vdots$
    >   - Calibrating $\cdots$

61. L 471: "potential application of it" ⟹ "'potential application of this study" (Please describe 'it' explicitly. Please check if 'it' can be replaced with 'this study'; otherwise, please describe it adequately.)

62. L 472: "adopting it" ⟶ Again, describe 'it' explicitly.

---

## Author Response (AR1)

First, we appreciate the reviewers taking the time and effort to evaluate our research and we are pleased to receive the constructive and helpful feedback and comments. We have carefully considered all suggestions and comments raised by the reviewers and provided our explanations and modified the manuscript accordingly. Point-by-point responses to the Reviewers' comments are as follows. And the relevant manuscripts of this revision are pasted in dark green text, with red color for addition and light blue for deletion. We sincerely hope that the revised manuscript is now suitable for publication in the Geoscientific Model Development.

**Response to Referee #1**

Implementations of realistic crop managements are crucial for current land surface models to better simulate crop growth simulations. This manuscript applied double-cropping and interactive irrigation in Noah-MP that better captured crop growth seasonal variations in the North China Plain. The manuscript is well written and the overall flow is clear. There were several comments need to be addressed before considering for publication.

► We appreciate the reviewer's positive feedback and helpful comments. Please find our detailed responses to each comment below.

Comments:

1. The authors mentioned incorporating crop growth is important for understanding the land-atmosphere interactions, but the manuscript did not show any land surface simulations. I suggest the authors include a result section to show how the energy fluxes and soil moisture changed with double-cropping and irrigation.

► Thanks for the constructive comments, we add a figure (Figure 6 in the revised manuscript) to show the spatial and monthly changes induced by double-cropping and irrigation.

Line 382 in Section 3.1:
"Figure 6 presents irrigation impact on energy partition by depicting the differences between the irrigation simulation (IRRnew) and the non-irrigation simulation (CROPnew). The upper panel visualizes the spatial changes, while the lower panel illustrates the monthly averaged changes for the entire NCP region (represented by the blue line) and the double-cropping region (represented by the orange line). As expected, the increased soil moisture contributes to a higher latent heat flux, with maximum increase over 40 W/m². Conversely, irrigation-induced evaporation cools the surface, leading to a reduction in sensible heat flux, with the sharpest decrease around 30 W/m². The cooler surface also reduced longwave radiation emitted from the surface, causing increases in net radiation with the greatest change about 15 W/m². Overall, the increase in latent heat flux surpasses the decrease in sensible heat flux, and when combined, their changes partially balance out to equal the net radiation. The most substantial changes are observed in southern Hebei province, which aligns with the irrigation fraction map (Fig. 4c). In the lower panel, all monthly patterns exhibit two peaks, with a larger peak in June and a smaller peak in September. The monthly pattern within the double-cropping area shows more pronounced changes and a more distinct two-peak structure. Furthermore, the irrigation responses of all variables display similar spatial and temporal patterns to the irrigation amount, indicating a strong correlation between irrigation application and these observed changes."

[Figure]

Figure 6. Irrigation-induced changes (IRRnew-CROPnew) in the climatology spatial pattern (upper panel) and mean monthly pattern (lower panel) of various variables, including irrigation, soil moisture, latent heat flux, sensible heat flux, and net radiation. The blue line represents the average value for all grids in the North China Plain (NCP), while the orange lines correspond to the double-cropping area only.

2. Although the recalibration process was described in the supplement, it still unclear whether such calibration of the double cropping scheme only at Yucheng site then applied the calibrated parameters across the whole domain? Or you perform calibration at Yucheng site and at regional scale separately?

► Thanks for the question. Yes, we conducted the calibration process initially at the Yucheng site, as demonstrated in Fig. 7. Next, we applied the calibrated parameters to the entire study domain. However, to ensure the accuracy and reliability of our model, we performed validation by assessing the spatial patterns of various variables, including crop calendar, grain mass, leaf area index (LAI), and vegetation fraction (FVEG). To address any potential ambiguity, we have included a clear explanation in the revised manuscript.

**Line 261 in Section 2.3.3:**
Table S1 provides the adjusted parameters for wheat and maize, along with the supporting scientific references. initially recalibrated in Yucheng and Shenyang using station data. Subsequently, these parameters are applied to the whole domain, with validation of vegetation pattern (i.e., LAI, FVEG, grain mass and crop calendar) conducted to ensure their spatial applicability to the whole region.

**Figure 3.** Spatial distribution of (a) the cropping system with two stations used for calibration, (b-e) harvest date and planting date for wheat and maize over a year based on the chronological order. 'E. Apr' and 'L. Apr' is the abbreviation for Early and Late April.

3. The analysis focused on the North China Plain, but all the plots showed much large region of eastern China that seems very distractive for me. I suggest to zoom in the North China Plain for all your plots.

► Thanks for the suggestion. We have updated the Figure 7 and 8 (labelled as Figure 8 and 9 in the revised manuscript) to zoom in the NCP region. For Figure 3 and Figure 4, we think it would be better to show the whole domain so it can display the provincial differences.

[Figure]

**Figure 8.** Comparison of the crop growth calendar and yield by comparing the heading date, maturity date, and annual yield for wheat and maize between observation (OBS) and simulation (CROPdef using default crop model, CROPnew using improved crop model, and IRRnew using both improved crop model and improved irrigation model).

[Figure]

**Figure 9.** Monthly LAI patterns of satellite observation (OBS), simulation with default crop model only (CROPdef), and simulation with improved crop and improved irrigation (IRRnew) from March (MAR) to October (OCT).

4. The current flow of section 3.2 is a little bit confusing. Why only show regional crop grain yield in 2005? The validations should focus on 2005-2014. I suggest to first show the site calibration (2005) and validation (2005-2014). Then show the regional crop calendar and grain yield validation for 2005-2014.

► We appreciate the suggestion and have made the necessary adjustments. We have extended the validation period for the crop calendar to a 10-year period as recommended. However, we would like to clarify that the available observation yield data (2015+_GAEZ) is limited to the year 2015. Consequently, the validation process for grain mass is restricted to comparing the 2015 observations with the simulation average from 2005 to 2014. We acknowledge this limitation and have mentioned it in the revised manuscript.

**Line 423 in Section 3.2.2:**
**3.2.2 Validation of crop calendar and grain mass**
To evaluate the performance of the stage identification process within the crop model, we compare the 10-year mean heading and maturity dates from each simulation with the satellite …

**Line 446 in Section 3.2.2:**
Due to the limited availability of grid-scale yield data, the computed 2015 crop yield from the Global Agro-Ecological Zones (GAEZ) model is used as the observational benchmark …

5. Figure 8. What LAI data used in the validation?

► Thanks for the question. We used a reprocessed MODIS data from Sun Yat-sen University, and we have added the information in the data availability statement.

**Line 556 in Data Availability Statement:**
LAI dataset is initially Sun Yat-sen University (Yuan et al., 2020), shown at http://globalchange.bnu.edu.cn/data/global_lai_0.1/.

Yuan, H., Dai, Y., and Li, S.: Reprocessed MODIS Version 6 Leaf Area Index data sets for land surface and climate modelling., 2020.

6. Line 22-24. I don't think it is necessary to list the next phase of your research in the abstract.

► Thanks for the comment. We have deleted it accordingly, and put some implications of the study instead.

**Line 21 in Abstract:**
 The improved simulation for large-scale irrigated crop in the NCP can further enhance our understanding of the intricate relationship between agricultural development and climate change…

7. Line 120. The WRF model version is 4.3, but it was 4.5 in your title.

► It should be WRF4.5. We have updated it in the revised manuscript.

**Line 132 in Section 2.2:**
The study employs the Advanced Research version of the WRF Model (version 4.5)

8. The experiment design did not mention the nesting and the domain range. The figure 1 made me think you have double-nesting domains, where the smaller rectangle is the inner domain.

► We only did one-domain simulation. The smaller rectangle just highlight the NCP region that we would like to pay more attention to. We show the whole picture since we want to apply this double-cropped irrigation to study the irrigation impact in local as well as its surrounding region, so we want to make sure the model also make reasonable performance in the whole region. We added some explicit explanation in the experiment design part to avoid misunderstanding.

Our study focused on conducting a one-domain simulation that covers the whole eastern China, with the highlighted smaller rectangle representing the NCP region of particular interest. While our primary focus was the NCP, we also aimed to assess the model's performance in the surrounding areas. Therefore, we initially presented the model performance in the whole eastern China. To prevent any potential misunderstandings, we have included explicit explanations in the experiment design section, providing further clarity on our intentions.

**Line 133 in Section 2.2:**
Model domain is shown in Fig. 1. This study only employs a single domain which is depicted as the entire map in Fig. 1, while the inner black box in Fig. 1 serves solely for the identification of the NCP region.

9. I suggest to rearrange figure 4. The color bar for irrigation fraction plot is under the targe irrigation plot. Similar problem for figure 6, you want to move the green-blue color bar up so it could underneath the maturity plots.

►Thanks for the comments. Figure 4 have been updated accordingly. But Figure 6 (or Figure 7 in revised manuscript), as the second reviewer suggested, is rearranged according to the order that they appear in the explanation. But we have include separated colorbars beneath each subplot to improve clarity. We have also explicitly stated that all "blue" subplots actually share the same color scale in the figure caption.

[Figure]

**Figure 4.** Spatial maps of 2005 (a) agricultural usage, (b) estimated irrigation usage, (c) irrigation fraction (same as Fig. 1d), (d) statistical irrigation, e) satellite irrigation, f) observation irrigation, (g) simulated irrigation using the default irrigation scheme (IRRdef), (h) simulated irrigation using improved irrigation scheme (IRRnew), (i) MAD (Manageable allowable deficit) irrigation threshold adopted in IRRnew and (j) irrigation range among 10 ensemble members using different initial conditions. For easy comparison, all subplots with blue colors (Fig. 4a,b,d,e,f,g,h) adopt the same color scale.

**Response to Referee #2**

Recommendation: Accept with minor revision.

This study describes the usefulness of the WRF-Crop model in capturing vegetation and irrigation patterns in the North China Plain (NCP), by incorporating double-cropping with interactive irrigation. The authors modified the crop model in terms of the vegetation fraction (FVEG) and the planting/harvesting dates and improved the irrigation model in calculating the irrigation amount. The authors validated their model results in terms of various irrigation and crop growth aspects and concluded that coupling of the enhanced crop and irrigation models significantly improved the performance in estimating crop stages and yields, field biomass, and leaf area index. This manuscript can be a valuable report to the scientific community for better prediction of double cropping and irrigation aspects in NCP; however, some issues need to be clarified or discussed in more detail.

► We appreciate the reviewer's positive feedback and helpful comments. Please find our detailed responses to each comment below.

1. The authors need to specifiy major differences in their methods and results compared with those of Yu et al. (2022).

► Thank you for the insightful comment. Yu et al. (2022) focus on northeast China, which predominantly practices single-cropping and rainfed cultivation. They utilize the original single-cropping model without irrigation. However, our study focus on the irrigated North China Plain (NCP), where both single and double cropping systems are prevalent, with double cropping being more dominant. Thus, we implement a double-cropping function with activated irrigation to better represent the NCP's agricultural practices. Detailed comparison are listed as follows:

Calibration Process: While Yu et al. (2022) mainly calibrate the specific leaf area (BIO2LAI) for one-year corn or soybean that already present in WRF4.2, we develop almost a full set of parameters for winter wheat and recalibrate the carbon allocation for summer corn, which differs significantly from one-year corn.

Phenology Input: Yu et al. (2022) use unified planting and harvesting dates across their study domain. In contrast, we adopt spatially varied dates to better capture the diverse agricultural practices within our study area.

Results: Yu et al. (2022) evaluate the performance of their dynamic crop model at a fine resolution (10 km) in northeast China, showing variability in results with different parameters and comparing vegetation predictions from crop and dynamic vegetation models. Our study, on the other hand, aims to develop an irrigated crop model applicable to irrigation studies. Thus, we use a 25 km resolution across eastern China, focusing on the model's ability to capture the general patterns of both water and vegetation, particularly the vegetation's sensitivity to water forcing (i.e., comparing irrigation and non-irrigation scenarios).

Due to the adoption of different study domains, it is challenging to conduct a result comparison quantitatively. However, we believe that distinctions mentioned above already highlight the unique contributions of our study in advancing the understanding of irrigated cropping systems in the NCP. In conclusion, we only adopted the method of calibrating BIO2LAI and utilized the default planting and harvesting dates (for a few grids that are not covered by the crop calendar dataset) from Yu et al. (2022). Thus, it's difficult to incorporate a detailed comparison into our manuscript, as the primary improvements we have made, mainly the implementation of double-cropping rotation and spatially-varied interactive irrigation, are not addressed in Yu et al. (2022). Nevertheless, we have added a reference to Yu et al. (2022) in the introduction section to acknowledge the similar parameter calibration process carried out in their study.

Yu, L., Liu, Y., Liu, T., Yu, E., Bu, K., Jia, Q., Shen, L., Zheng, X., and Zhang, S.: Coupling localized Noah-MP-Crop model with the WRF model improved dynamic crop growth simulation across Northeast China, Comput. Electron. Agric., 201, 107323, https://doi.org/10.1016/j.compag.2022.107323, 2022.

**Line 72 in Section 1:**
Previous studies have shown that the regionalization process significantly improves the model performance. This process includes not only parameter calibration (Hong et al., 2015; Liu et al., 2010; Park and Park, 2021; Xie et al., 2007) but also algorithm modifications to enhance the model's applicability to different regions (Bou-Zeid et al., 2007; Livneh and Lettenmaier, 2013; Song et al., 2022) … For instance, recalibration has been shown to significantly enhance crop prediction accuracy in Northeast China and southwestern Europe (Asmus et al., 2023; Yu et al., 2022) …

**Line 172 in Section 2.3.1:**
A crop grid is defined based on MODIS land-use classification as either 'Croplands' or 'Cropland/Natural Vegetation Mosaic'. This definition aligns with Fan et al. (2023), and is similar to the approach employed by Yu et al. (2022) who set a threshold of 50% cropland percentage, since the majority of grids in the NCP region contain over 90% cropland (Fig. 1b).

**Line 231 in Section 2.3.2:**
Few grids not covered by the satellite dataset are assigned 1 May (121st Julian Day) and 11 October (284th Julian Day) as the default planting and harvesting date for maize, respectively, based on field study (Yu et al., 2022).

**Line 279 in Section 2.3.3:**
For maize, the values of VCMX25 and AVCMX have simply followed the previous studies, while BIO2LAI is subject to recalibration, as its necessity of recalibration has been demonstrated by Yu et al. (2022).

2. Abstract: Delete the last sentence describing the future research. Just include more details and focus on the current research.

► Thanks for the comment. We have deleted it accordingly, and put some implications of the study instead.

**Line 21 in Abstract:**
 The improved simulation for large-scale irrigated crop in the NCP can further enhance our understanding of the intricate relationship between agricultural development and climate change…

3. Plain Language Summary: This part should have more scientific information, including more details in results and their implications.

► Thanks for the comment. However, GMD limits the length as 500 characters with space, which makes it very difficult to provide more details. But still, we have tried our best to rewrite it to include as much details as we can.

**Plain Language Summary:**

Irrigated agriculture in the North China Plain (NCP) has a significant impact on the local climate, but the existing climate models do not accurately simulate the crop and irrigation. To address this limitation, we add a double-cropping function to Weather Research Forecast model. We also recalibrate the parameters to simulate the crop growth and irrigation amounts more accurately. Our improved model better captures crop calendar, biomass, vegetation fraction, as well as monthly leaf area index.

4. L 53 & L 78–79: Remove the commas in front of 'but alo'. Note that 'not only . . . but also' requires a comma only when two independent clauses are linked. No comma is required in linking nouns or noun phrases.

► They have been updated accordingly.

**Line 51 in Section 1:**

… these interactive crop models can not only capture the temporal pattern of crop growth, but also depict spatial heteroneity at regional scales with relatively fast computational speed …

**Line 89 in Section 1:**

… the NCP is an ideal testbed for studying irrigated crops and climate feedback, rooting not only in its extensive cropland and high productivity, but also in its semi-arid background, and intense irrigation …

5. L55: "while others incorporate irrigation with fixed amount (Vira et al., 2019) or dynamically based on daily soil conditions" −→ Hard to understand: Rewrite. Should 'dynamically' be replaced with 'dynamically varying amount' or something else?

► Yes, 'dynamically' implies the irrigation amount is varied dynamically with the soil conditions instead of a fixed value. We have revised it accordingly.

**Line 54 in Section 1:**
When simulating the water forcings that sustain crop growth, some models simply assume no irrigation (Liu et al., 2016), while others incorporate irrigation with fixed amount (Vira et al., 2019) or dynamically adjust the irrigation amount based on daily soil conditions

6. L 69: "regionalizing the algorithms" −→ How can the algorithms be regionalized? In general, one develops new parameterization schemes or improves the existing parameterization schemes and/or tunes/calibrates/optimizes the parameter values in the schemes, making the schemes and/or parameter values work well in a specific region of interest. The authors need to explain more explicityly on 'regionalization of algorithms'.

► In this study, the process of regionalization not only includes parameter calibration but also some specific modifications on the algorithms. For example, we introduced a temperature check for irrigation to avoid harmful irrigation during freezing periods. This modification may not be necessary in other regions where double cropping is not dominant (i.e., winter is not a cropping period for them). Thus, we also consider these modifications as part of the regionalization process, as they mainly aim to improve the model's suitability for the targeted region. We have included some additional explanations in the revised manuscript to avoid confusion.

**Line in 69 Section 1:**

… regionalizing the  model and improving their adaptation for large-scale irrigation over other parts of the world becomes imperative …

7. L 71: Include a paragraph that provides examples of 'regionalizing algorithms', e.g., developing new algorithms for a specific region and tuning/calibrating/optimizing parameter values that fits observations well in a specific region. Some examples of developing parameterizatin schemes at regional scales, in chronical order, include

Bou-Zeid, E., Parlange, M. B., and Meneveau, C.: On the Parameterization of Surface Roughness at Regional Scales. J. Atmos. Sci., 64, 216–227, https://doi.org/10.1175/JAS3826.1, 2007.

Liu, J., Ding, Y., Zhou, X., Li, Y.: A Parameterization Scheme for Regional Average Runoff over Heterogeneous Land Surface Under Climatic Rainfall Forcing, J. Meteorol. Res., 24, 116-122, 2010.

Song, W., Tang, H., Sun, X., Xiang, Y., Ma, X., Zhang, H.: Developing a New ParameterizationScheme of Temperature Lapse Ratefor the Hydrological Simulation ina Glacierized Basin Based on Remote Sensing. Remote Sens., 14, 4973, https://doi.org/10.3390/rs14194973, 2022.

Asmus, C., Hoffmann, P., Pietik̈ainen, J.-P., B̈ohner, J., and Rechid, D.: Modeling and evaluating the effects of irrigation on land–atmosphere interaction in southwestern Europe with the regional climate model REMO2020–iMOVE using a newly developed parameterization, Geosci. Model Dev., 16, 7311–7337, https://doi.org/10.5194/gmd-16-7311-2023, 2023.

and some examples of regional parameter estimations, in chronical order, include

Xie, Z., Yuan, F., Duan, Q., Zheng, J., Liang, M., and Chen, F.: Regional Parameter Estimation of the VIC Land Surface Model: Methodology and Application to River Basins in China. J. Hydrometeor.,8, 447–468, https://doi.org/10.1175/JHM568.1, 2007.

Livneh, B., and Lettenmaier, D. P.: Regional parameter estimation for the unified land model, Water Resour. Res., 49, https://doi.org/10.1029/2012WR012220, 2013.

Hong, S., Park, S. K., and Yu, X.: Scheme-Based Optimization of Land Surface Model Using a Micro-Genetic Algorithm: Assessment of Its Performance and Usability for Regional Applications, SOLA, 11, 129-133, https://doi.org/10.2151/sola.2015-030, 2015.

Park, S. and Park, S. K.: A micro-genetic algorithm (GA v1.7.1a) for combinatorial optimization of physics parameterizations in the Weather Research and Forecasting model (v4.0.3) for quantitative precipitation forecast in Korea, Geosci. Model Dev., 14, 6241–6255, https://doi.org/10.5194/gmd-14-6241-2021, 2021.

► We sincerely appreciate the constructive comments and the detailed references provided by the reviewer. While these references are applicable to the context of the general regionalization process, some of them appear to be less relevant to our focus on irrigation and crop simulations. Therefore, we incorporate these references in the beginning of the paragraph when addressing the general aspects of regionalization, and then provide more detailed explanations

for the studies that are pertinent to our research. The revised paragraph is highlighted in red text below. Once again, we extend our gratitude to the reviewer for the thorough suggestions on potential references.

**Line 72 in Section 1:**

Previous studies have shown that the regionalization process significantly improves the model performance. This process includes not only parameter calibration (Hong et al., 2015; Liu et al., 2010; Park and Park, 2021; Xie et al., 2007) but also algorithm modifications to enhance the model's applicability to different regions (Bou-Zeid et al., 2007; Livneh and Lettenmaier, 2013; Song et al., 2022). For instance, recalibration has been shown to significantly enhance crop prediction accuracy in Northeast China and southwestern Europe (Asmus et al., 2023; Yu et al., 2022). Introducing new tuning factors into the default equation can aid in simulating unique vegetation patterns within specific study domains (Wu et al., 2018b). Upgrading a variable such as the irrigation threshold from a single constant to a spatially varied 2D variable can better capture the spatial variability of irrigation application (Xu et al., 2019; Zhang et al., 2020). Additionally, incorporating new irrigation methods for paddy cropland improved irrigation predictions for southern Asia (Yao et al., 2022). These enhancements underscore the importance and efficacy of regionalization in improving the simulation in irrigated agriculture.

8. L 84: The acronym 'LSM' should be replaced with 'LSMs'.

► It has been updated accordingly.

**Line 95 in Section 1:**

… most current crop models in land surface models (LSMs) primarily account for single cropping

9. L 89–90: "Noah-Multiparameterization (Noah-MP)" −→ Give the full original words "Noah land surface model with multiparameterization options (Noah-MP)", and provide a proper reference for it, i.e., Niu et al. (2011).

► It has been updated accordingly.

**Line 100 in Section 1:**

… Noah  land surface model with multiparameterization options (Noah-MP) (Niu et al., 2011) has been selected, as it already encompasses several functions related to cultivation simulation, and has consistently exhibited exemplary performance in previous studies when simulating single-cropping scenarios …
10. L 92–93: "Weather Research Forecast (WRF)" =⇒ "Weather Research and Forecasting (WRF) model", i.e., provide the correct original words for WRF. And cite a proper reference.

► It has been updated accordingly.

**Line 103 in Section 1:**

Its crop model is already implemented within the Weather Research and Forecasting Model (WRF) (Skamarock et al., 2019) to enable two-way nested feedback between the crop system and climate dynamics …

11. Figure 1: Modify the caption to "(a) Annual precipitation (mm/day) and basic geostatic variables applied in this project, including (b) topography (m), (c) cropland fraction (%), and (d) irrigated land fraction (%)."

► Thanks for the comment. We also found these variables are disordered. The order has been also corrected in the revised manuscript.

**Figure 1:** (a) Topography (m), (b) Cropland fraction (%), (c) annual precipitation (mm/day), and (d) irrigated land fraction (%).

12. L 120: Provide the proper reference of WRF version 4.3. Please also check if this version number is correct because your title says it is WRF4.5.

► Thanks for pointing out the typo. The correct version should indeed be WRF 4.5. We have made the necessary correction to ensure accuracy in our paper.

**Line 132 in Section 2.2:**
The study employs the Advanced Research version of the WRF Model (version 4.5)

13. L 127: "ERA5-Interim" −→There is no ERA5-Interim reanalysis data. It should be either "ERA5" or "ERA-Interim". Based on the cited reference, Hersbach et al. (2020), it should be "ERA5".

► We conducted preliminary tests using both ERA5 and ERA-Interim datasets, and based on our evaluation, ERA-Interim performed better. Thus, it should be ERA-Interim. The name and the reference have been corrected.

**Line 141 in Section 2.2:**
… initial and lateral boundary conditions are obtained from the 6-hourly ERA-Interim reanalysis dataset, which helps to reduce the uncertainty arising from the boundary condition ( Dee et al., 2011)…

14. L 143: There exists only one Liu et al. (2016) and only one Zhang et al. (2020) in the References; thus, the authors do not need to put the first names' initials. Replace "X. Liu et al. (2016)" with "Liu et al. (2016)" and "Z. Zhang et al. (2020)" with "Zhang et al. (2020)". Please do it throughout the manuscript.

► It has been updated accordingly.

**Line 155 in Section 2.2:**
… default crop model is conducted using the original scheme proposed by  Liu et al. (2016) and parameters derived from  Zhang et al. (2020) …

**Line 169 in Section 2.3.1:**
… the crop module developed by Liu et al. (2016) is selected as the foundation for crop simulation …

15. L 175: There exists only one Wu et al. (2018b) in the References; thus, replace "L. Wu et al. (2018b)" with "Wu et al. (2018b)".

► It has been updated accordingly.

**Line 187 in Section 3.1:**
… equation is proposed by Niu et al. (2011) and further testified by  Wu et al. (2018b) in the NCP region …

16. L 194: "crop growth." =⇒ "crop growth. Equations (1) and (2) below represent the original FVEG equation by Niu et al. (2011) and the adjusted FVEG suggested in this study, respectively:"

► It has been updated accordingly.

**Line 207 in Section 3.1:**
… and an underestimation towards the later stages of crop growth. Equations (1) and (2) below represent the original FVEG equation by Niu et al. (2011) and the adjusted FVEG suggested in this study, respectively: …

17. L 196, Eq. (1): "Niu et al. (2011) FVEG" =⇒ "Original FVEG"

► It has been updated accordingly.

**Line 210 in Section 3.1:**
 Original $\text{FVEG} = 1 - e^{(-0.52 \times (\text{LAI} + \text{SAI}))}$ , $\text{FVEG} \in [0,1]$

18. L 222–223: Describe explicity 'Yucheng' and 'Shenyang' in Fig. 3a. In Fig. 3a, two red circles may represent these two locations, but the authors needs to put the location names in the map. Most readers around the world are not familiar with the location names.

► Thanks for the comment. It has been updated accordingly.

[Figure]

**Figure 3.** Spatial distribution of (a) the cropping system with two stations used for calibration, (b-e) harvest date and planting date for wheat and maize over a year based on the chronological order. 'E. Apr' and 'L. Apr' is the abbreviation for Early and Late April.

19. L 225: The double-cropping area is quite large: Would Yucheng be considered to represent well the large area, especially the southern part of the area? Isn't there any station near the southern edge of the area? If another station exists with long-term observation at the southern part of the area, including it will make this study more valuable. Otherwise, add a sentence that Yucheng, located at the northern part of the double-cropping area, can well represent the characteristics of the southern part of the area, with some scientific evidences.

► Thanks for the question. We did calibration first in the Yucheng site (shown in Fig. 7), and then we applied it to the whole domain. Thus, the performance of adopting Yucheng in other area can be examined by validating the spatial pattern of crop calendar, grain mass, LAI and FVEG (Fig. 8 and 9 and S2). Please refer to the response to comment #59 for a detailed explanation regarding the choice of Yucheng over other stations in the southern region.

20. L 231: The authors used 'Vcmx25' and 'BIO2LAI' without any definition or explanation. If they are acronyms, please provide the original worrds; otherwise, explain what they are.

► The original words are provided inline.

**Line 254 in Section 2.3.3:**
For instance, large regional uncertainties may exist in **the rubisco capacity** (Vcmx25) and **the leaf area per living leaf biomass** (BIO2LAI) for summer maize…"

21. L 237: "Table S1 provides . . . with the supporting scientific references and recalibration procedures." =⇒ "Table S1 provides . . . with the supporting scientific references." Then, from Supplementary, please move the 2 paragraphs below "The recalibration process:" to the end of Section 2.3.3.

► It has been updated accordingly.

**Line 261 in Section 2.3.3:**
Table S1 provides the adjusted parameters for wheat and maize, along with the supporting scientific references . Parameters are initially recalibrated in Yucheng and Shenyang using station data. Subsequently, these parameters are applied to the whole domain, with validation of vegetation pattern (i.e., LAI, FVEG, grain mass and crop calendar) conducted to ensure their spatial applicability to the whole region.

The recalibration starts from crop-stage identification, since it relies purely on the accumulated GDD and is less affected by other crop parameters. The GDD-related parameters are retrieved from Zhang et al. (2020) and Zhang et al. (1991), and then validated with the heading date and maturity date retrieved from the satellite data (Luo et al., 2020). The crop stage comprises the pre-planting stage, three vegetative stages (emergence, initial vegetative, post-vegetative), two reproductive stages (initial reproductive, post-reproductive), and finally, one maturity stage. During the vegetative stage, a majority of carbohydrates are allocated to the leaves and stems, while in the reproductive stage, the allocation shifts towards the grain.

Next, the general growth rate including BIO2LAI can be extracted from the station data, and Vcmx25 can also be estimated using the monthly satellite data of gross primary product (GPP) and LAI, since the photosynthesis rate and the LAI can be considered linearly related, especially on sunny days when the canopy temperature is around 25°C (He et al., 2023). The GPP and LAI that we adopted for validation are initially derived from MODIS products but have undergone further post-processing to generate a more continuous monthly pattern (Wang et al., 2020; Yuan et al.,

2020). Furthermore, the AVCMX, which represents the crop sensitivity to the temperature, can be determined by the gradient of biomass accumulation (Huang et al., 2022), especially in spring and autumn with greater temperature changes. For maize, the values of VCMX25 and AVCMX have simply followed the previous studies, while BIO2LAI is subject to recalibration, as its necessity of recalibration has been demonstrated by Yu et al. (2022).

Following the establishment of the general photosynthesis rate, we proceed to fine-tune the distribution of carbohydrates among the leaf, stem, and grain compartments, based on the annual cycle of leaf mass and stem data obtained from the station data. Any remaining carbohydrates are allocated to the root. In cases where the recalibration of the distribution scheme alone does not yield satisfactory predictions, adjustments to the turnover and translocation rates are implemented. Additionally, the crop yield will be validated through comparisons with remotely sensed estimations from Grogan et al. (2022).

22. Figure 3: In the caption of Fig. 3, explain what the two red circles in Fig. 3a means.

► It has been updated accordingly. And the name of those stations are labelled in the figure.

**Figure 3**. Spatial distribution of (a) the cropping system with two stations used for calibration, (b-e) harvest date and planting date for wheat and maize over a year based on the chronological order. 'E. Apr' and 'L. Apr' is the abbreviation for Early and Late April.

23. Figure 3: Figures 3b-e are never cited in the manucript. The authors need to properly cite each subfigure in the text, probably in the paragraph in L 210–219.

►Thanks for the comment. The corresponding explanation to Figures 3b-e has been added in section 2.3.3, after the definition of cropping area.

**Line 261 in Section 2.3.3:**
Based on the defined cropping area, the planting and harvesting dates are determined using the method outlined in section 2.3.2. The chronological sequence of these dates is presented in Figs. 3b-e. In regions with a single cropping system, spring maize is typically planted in May and harvested in September. On the other hand, in those double-cropping regions, winter wheat is usually harvested in late May or early June, immediately followed by the planting of summer maize. Next, maize harvest generally takes place in late September or early October, again followed by the planting of winter wheat, which continues to grow until the next year.

24. L 249: Use consistent tense: "the crop emerges" vs. "the crop matured".

► It has been updated to the present tense.

**Line 297 in Section 2.4:**
… irrigation is initiated when the crop emerges and stopped when the crop physiologically matures…

25. L 254: Is there any reason that 'irrigation' should be expressed in capital ("The default Irrigation")?

► It's just a typo, thanks for pointing out it. We have corrected it accordingly.

**Line 302 in Section 2.4:**
The default irrigation can be activated anytime …

26. L 267: What does the number '2005' mean? Is it the year 2005?
► Yes. We have updated it to 'year 2005' to avoid confusion.

**Line 320 in Section 2.4:**
the irrigation land fraction map around year 2005 from the Food Agriculture Organization database

27. Equations 3 & 4: Define SMCLIM and SMCAV L immediately following the equations.
► Please refer to Comment #29.

28. L 264–269: It is recommended to modify this part as below: The default daily irrigation amount is resolved as follows, based on the soil moisture and vegetation fraction which is fixed to be 0.95:

$$\int (\text{SMCLIM} - \text{SMCAVL}) * 0.95$$

where SMCLIM is ... and SMCAV L is .... When adopting it to large-scale irrigation, we improve the irrigation amount by replacing the constant 0.95 with IRRFRA, i.e., the irrigation land fraction map around 2005 from the Food Agriculture Organization database (Siebert et al., 2013) as follows

$$\int (\text{SMCLIM} - \text{SMCAVL}) * \text{IRRFRA}$$

► Please refer to Comment #29.

29. L 272–273: Definition of SMCLIM and SMCAVL should appear immediately following Equations 3 & 4.

► We have made updates accordingly. However, we moved the explanation of SMCLIM and SMCAVL before the equations for better explanation.

**Line 312 in Section 2.4:**
Irrigation is required when the soil moisture is lower than the predefined irrigation threshold called management allowable deficit (MAD). MAD is a decimal number between 0 and 1, indicating the cursor between the wilting and the saturated soil moisture.  The expected soil moisture after irrigation (SMCLIM) is defined by the MAD, and the soil water deficit is the gap between current soil moisture availability (SMCAVL) and SMCLIM. The total irrigation amount is the integrated deficit of all soil layers. Thus, the default daily irrigation amount is resolved as follows, based on the soil moisture and vegetation fraction which is fixed to be 0.95:

$$\int (\text{SMCLIM} - \text{SMCAVL}) * 0.95$$

When adopting it to large-scale irrigation, we improve the irrigation amount by replacing the constant 0.with IRRFRA, i.e., the irrigation land fraction map around year 2005 from the Food Agriculture Organization database (Siebert et al., 2013) as follows

$$\int (SMCLIM - SMCAVL) * IRRFRA$$

It is also stated that the county-level calibrated MAD significantly enhances the irrigation prediction (Xu et al., 2019; Zhang et al., 2020). Similarly, we calibrated the irrigation threshold province by province using the updated irrigation function, and finally apply this MAD spatial map to IRRnew. As a comparison, IRRdef only adopts 0.8 as a uniform threshold which is simply calibrated by the national total amount (Fan et al., 2023).

30. L 284: "which not only comprises irrigation, but also husbandry, forestry, and fishery consumption" =⇒ "which comprises not only irrigation but also husbandry, forestry, and fishery consumption"

► It has been updated accordingly.

**Line 332 in Section 3.1:**
it is only provided as annual agricultural water usage which not only comprises irrigation, but also husbandry, forestry, and fishery consumption

31. L 285–287: Figures 4b and 4c are cited before Fig. 4a is cited. It is recommended to switch the order of subfigures in the order that they appear in the explanation.

► It has been updated accordingly.

[Figure]

**Figure 4.** Spatial maps of 2005 (a) agricultural usage, (b) estimated irrigation usage, (c) irrigation fraction (same as Fig. 1d), (d) statistical irrigation, e) satellite irrigation, f) observation irrigation, (g) simulated irrigation using the default irrigation scheme (IRRdef), (h) simulated irrigation using improved irrigation scheme (IRRnew),  (i) MAD (Manageable allowable deficit) irrigation threshold adopted in IRRnew and (j) irrigation range among 10 ensemble members using different initial conditions. For easy comparison, all subplots with blue colors (Fig. 4a,b,d,e,f,g,h) adopt the same color scale.

32. L 296–297: "NCP (i.e., Beijing, Tianjin, Hebei, Shandong, and Henan, follows D. Wu et al., 2018) is coupled" =⇒ "NCP—Beijing, Tianjin, Hebei, Shandong and Henan that follow Wu et al. (2018)—is coupled"

► It has been updated accordingly.

**Line 345 in Section 3.1:** Beijing, Tianjin, Hebei, Shandong, and Henan, follows Wu et al., 2018a)

33. L 307: "Figure 4(j)" =⇒ "Figure 4j"

► It has been updated accordingly.

**Line 352 in Section 3.1:** Figure 44i presents the province-based MAD threshold…

34. L 307–315: Figure 4j is cited before Figure 4i is cited: Switch the order of these two subfigures in Fig. 4.

► It has been updated accordingly. Please refer to comment #31 for the revised figure.

35. Figure 4, Caption: "(IRRnew), and (i) irrigation range among 10 ensemble members using different initial conditions (j) MAD" =⇒ "(IRRnew), (i) irrigation range among 10 ensemble members using different initial conditions, and (j) MAD"

► It has been updated accordingly. Please refer to comment #31 for the revised figure.

36. Figure 5 and L 319–329: The authors just compared the results between two models—IRRdef vs. IRRnew. To verify that IRRnew is better, the authos should also show the observations. Although the authors showed that IRRnew had better results than IRRdef in terms of spatial distributions in Fig. 4, the authors should also validate the model results in terms of temporal variations.

► Thanks for the constructive comments. We totally agree that temporal validation is also important, but the statistical data is provided annually but not monthly. Thus, we only assess the inter-annual variability of irrigation prediction in Figure 10.

37. L 337: "(first two lines in Fig. 6)" =⇒ "(top and middle panels in Fig. 6)"

► It has been updated accordingly. Please refer to comment #38 for the revised paragraph.

38. L 338: "we considered the entering the initial reproductive stage as the heading date" =⇒ "we regarded the start of the initial reproductive stage as the heading date"

► It has been updated accordingly.

**Line 324 in Section 3.2.2:**
To evaluate the performance of the stage identification process within the crop model, we compare the 10-year mean heading and maturity dates from each simulation with the satellite estimations ( top and middle panels in Fig. 8). Since the model accumulates carbon to grain starting from the initial reproductive stage, we  regarded the  start of the initial reproductive stage as the heading date, which aligns with the heading date identified by the time of maximum LAI in the satellite estimation. Similarly, the transition day from the post-reproductive stage to the maturity stage is  regarded as the maturity date. According to the algorithm, the heading and maturity dates can be  regarded as rough indicators of the transition from the vegetative stage to the reproductive stage, and ultimately to the maturity stage. This validation process allows us to assess the model's ability to accurately simulate the temporal development of crop growth.

39. L 340: "is considered as the maturity date" =⇒ "is considered the maturity date" or "is regarded as the maturity date"

► It has been updated accordingly. Please refer to comment #38 for the revised paragraph.

40. L 341: "can be considered as rough indicators" =⇒ "can be considered rough indicators" or "can be regarded as rough indicators"

► It has been updated accordingly. Please refer to comment #38 for the revised paragraph.

41. L 351: "time Is not" =⇒ "time is not"

► It has been updated accordingly.

**Line 437 in Section 3.2.2:**
This suggests that employing a uniform starting and ending time  is not suitable for a regional domain.

42. L 356: "(third row in Fig. 6)" =⇒ "(bottom panels in Fig. 6)"

► It has been updated accordingly.

**Line 446 in Section 3.2.2:**
Similar enhancement can be observed when assessing the crop yield ( bottom panels in Fig. 8).

43. L 365: "each of the following factors, implementation … irrigation, holds" =⇒"each of the following factors—implementation … irrigation— holds"

▶ It has been updated accordingly.

… each of the following factors,—implementation of double cropping, adoption of spatially varying input, and integration of irrigation—, holds significant importance in accurately simulating the crop calendar and grain yield …

44. L 368: "Yucheng and Shenyang station" =⇒ "Yucheng and Shenyang" (We already know that Yucheng and Shenyang are stations.)

▶ It has been updated accordingly.

**3.2.1 Validation of biomass in Yucheng and Shenyang **
45. L 369: "The station-based biomass is adopted for calibration (Fig. 7ad)." =⇒ "The station-based biomass in a specific year is adopted for calibration in both stations (Figs. 7a-d)."

▶ It has been updated as "The station-based biomass in year 2005 is adopted for calibration in both stations (Figs. 7a-d)."

The station-based biomass in year 2005 is adopted for calibration (Fig. 7a-d)

46. L 369 & 373: "Yucheng station" =⇒ "Yucheng"

▶ It has been updated accordingly.

… biomass cycle in Yucheng  clearly exhibits two distinct peaks, … does not make a noticeable impact at Shenyang  ((Figs. 7b and 7d). The long-term biomass results, displayed in Figs. 7e-g, … significant enhancements at the Yucheng  …

47. L 376: "Shenyang Station" =⇒ "Shenyang"

▶ It has been updated accordingly. Please refer to comment #46 for the revised sentence.

48. L 376: "(Fig. 7b and 7d)" =⇒ "(Figs. 7b and 7d)"

▶ It has been updated accordingly. Please refer to comment #46 for the revised sentence.

49. L 377: "Fig. 7e-g" =⇒ "Figs. 7e-g"

► It has been updated accordingly. Please refer to comment #46 for the revised sentence.

50. Figure S1: The bottom panels of Fig. S1 show "GWBnew", which is never defined either in the main text or in the Supplementary. Define "GWBnew".

► GWBnew is a redundant experiment of IRRnew. It has been removed in the updated supplementary.

51. Figure 8: The caption can be better described as follows: "Monthly LAI patterns of the satellite observation (OBS), simulation with default crop model only (CROPdef), and simulation with improved crop and improved irrigation (IRRnew) from March (MAR) to October (OCT)."

► It has been updated accordingly.

**Figure 9.** Monthly LAI patterns of satellite observation (OBS), simulation with default crop model only (CROPdef), and simulation with improved crop and improved irrigation (IRRnew) from March (MAR) to October (OCT).

52. L 417–418: "in the NCP, Shandong, Henan and Hebei, are depicted" =⇒ "in the NCP—Shandong, Henan and Hebei—are depicted"

► It has been updated accordingly.

**Line 489 in Section 3.2.4:**
Three provinces with large cropland extent in the NCP,—Shandong, Henan and Hebei,—are depicted in red dots with horizontal and vertical error bars showing the inter-annual variability of both observation and simulation, respectively. Most of the dots, especially the red dots, are located in …

53. L 418–419: "with horizontal and vertical error bars showing the interannual variability of both observation and simulation" =⇒ "with horizontal and vertical error bars showing the inter-annual variability of observation and simulation, respectively" (Please check if this modification gives correct interpretation.)

► Yes, it does. It has been updated accordingly. Please refer to comment #52 for the revised sentence.

54. L 419: "Most of the dots especially the red dots, are" =⇒ "Most of the dots, especially the red dots, are"

► It has been updated accordingly. Please refer to comment #52 for the revised sentence.

55. L 421: "the uncertainties associated with the observation and simulation are" =⇒ "the uncertainties associated with the observation and simulation, respectively, are"

► It has been updated accordingly.

**Line 493 in Section 3.2.4:**
… the uncertainties associated with the observation and simulation, respectively, are at least comparable …

56. Figure 9, Caption: "the horizontal and vertical error bars depict the inter-annual variability observed in both the simulations and actual measurements" =⇒ "the horizontal and vertical error bars depict the inter-annual variability observed in the simulations and the actual measurements, respectively"

► It has been updated as "horizontal and vertical error bars showing the inter-annual variability of observation and simulation, respectively"

**Figure 9**. Validation of the climatological mean of annual irrigation and crop yield across provinces. The red dots correspond to the three provinces with extensive cropland coverage in the North China Plain (NCP), while the horizontal and vertical error bars depict the inter-annual variability  of observation and simulation, respectively. The gray dots represent the remaining provinces.

57. L 430–431: "the model design restricts the simulation of only one crop type per grid" =⇒ "the model design restricts the simulation to only one crop type per grid"

► It has been updated accordingly.

**Line 501 in Section 4:** the model design restricts the simulation  to only one crop type per grid

58. L 432 & 433: "the Yucheng Station" =⇒ "Yucheng"

► It has been updated accordingly.

**Line 503 in Section 4:**
… summer maize at the Yucheng , which … leaf mass at the Yucheng  …

59. L 447: "It is important to acknowledge that the model performance may be less satisfactory in southern NCP."
−→ This may be because the authors did not include an observation station in the southern part of NCP for calibration (see the comment in #19). It will be great if the authors can include one station in the sourthern NCP for calibration and compare the results.

► Thanks for the comment. The reviewer highlighted an important issue. Yes, the model cannot simulate both the northern and southern regions well since they are sharing the same set of parameters, while certain parameters (such as specific leaf area) should differ regionally. This limitation, however, arises from model design that each crop type can only utilize one set of parameters for each crop type across the entire region. While there are other stations available in the southern NCP, the data collected from those stations indeed differ significantly from the Yucheng station. As a result, we were only able to choose one station for the calibration process, and we finally chose Yucheng due to the more intense irrigation in the northern NCP. We have explicitly mentioned this as the limitation of our study.

**Line 501 in Section 4:**
…, the model design restricts the simulation to only one crop type per grid. This simplification may contribute to inaccuracies in predicting the leaf mass of summer maize at the Yucheng, which can be revealed by the inconsistency of LAI observation (Fig. 9) in the NCP region and the leaf mass at the Yucheng (Fig. 7). While the LAI values indicate that September should have a smaller LAI compared to July (Fig. 9), the station data suggests that September actually has a greater leaf mass than July (Fig. 7). This discrepancy is likely attributed to two factors. Firstly, the specific leaf area, or BIO2LAI in the model, varies across different crop stages, as supported by both station data and existing literature (Amanullah, 2015; Zhou et al., 2020). In other words, the leaves may be thinner in July, while they become thicker in September. The second reason is that the observed LAI pattern represents a spatial average value over the grid, which may contain a diverse range of crops. Consequently, the specific station data for summer maize may not align well with the spatially averaged LAI. Since this study primarily focuses on the regional scale rather than individual field points, we prioritize matching the spatial LAI pattern while partially sacrificing the accuracy in predicting station biomass. As a result, the simulated LAI pattern is well-matched in the NCP region, while the predicted leaf mass for summer maize may not closely align with the station data. On the contrary, winter wheat greatly, even exclusively dominates the first crop season, and thus the station data and spatial pattern are consistent and can both be captured by the model (Fig. 7 and Fig. 9)…

60. L 453–454: This part should not be itemized, i.e., it should be rewritten as To enhance our understanding of the irrigation impact on regional climate, our study focuses on simulating irrigated crop growth in the NCP region using the WRF-Crop model. In order to improve the model's capabilities, we have implemented the following enhancements:
- Incorporating ...
- Establishing ...
...
- Calibrating ...

► It has been updated accordingly.

**Line 523 in Section 4:**
————To enhance our understanding of the irrigation impact on regional climate, our study …

61. L 471: "potential application of it" =⇒ "'potential application of this study" (Please describe 'it' explicitly. Please check if 'it' can be replaced with 'this study'; otherwise, please describe it adequately.)

► It represents the WRF model that this study adopts. It has been updated as "potential application of the WRF" in the revised manuscript.

**Line 541 in Section 4:**
… the potential application of  the WRF in other agricultural zones. And most of our validation data is derived from satellite observations, indicating the possibility of adopting  this model in regions even with limited ground-based data …

62. L 472: "adopting it" −→ Again, describe 'it' explicitly.

► Similarly, "it" has been updated as "this model". Please refer to comment #61 for the revised sentence.